# Projection Efficient Subgradient Method and Optimal Nonsmooth Frank-Wolfe Method

**Kiran Koshy Thekumparampil**
University of Illinois at Urbana-Champaign
thekump2@illinois.edu

**Prateek Jain**
Microsoft Research, India
prajain@microsoft.com

**Praneeth Netrapalli**
Microsoft Research, India
praneeth@microsoft.com

**Sewoong Oh**
University of Washington, Seattle
sewoong@cs.washington.edu

## Abstract

We consider the classical setting of optimizing a nonsmooth Lipschitz continuous convex function over a convex constraint set, when having access to a (stochastic) first-order oracle (FO) for the function and a projection oracle (PO) for the constraint set. It is well known that to achieve $\varepsilon$-suboptimality in high-dimensions, $\Theta(\varepsilon^{-2})$ FO calls are necessary [64]. This is achieved by the projected subgradient method (PGD) [11]. However, PGD also entails $\mathcal{O}(\varepsilon^{-2})$ PO calls, which may be computationally costlier than FO calls (e.g. nuclear norm constraints). Improving this PO calls complexity of PGD is largely unexplored, despite the fundamental nature of this problem and extensive literature. We present first such improvement. This only requires a mild assumption that the objective function, when extended to a slightly larger neighborhood of the constraint set, still remains Lipschitz and accessible via FO. In particular, we introduce MOPES method, which carefully combines Moreau-Yosida smoothing and accelerated first-order schemes. This is guaranteed to find a *feasible* $\varepsilon$-suboptimal solution using only $\mathcal{O}(\varepsilon^{-1})$ PO calls and optimal $\mathcal{O}(\varepsilon^{-2})$ FO calls. Further, instead of a PO if we only have a linear minimization oracle (LMO, à la Frank-Wolfe) to access the constraint set, an extension of our method, MOLES, finds a *feasible* $\varepsilon$-suboptimal solution using $\mathcal{O}(\varepsilon^{-2})$ LMO calls and FO calls—both match known lower bounds [54], resolving a question left open since [84]. Our experiments confirm that these methods achieve significant speedups over the state-of-the-art, for a problem with costly PO and LMO calls.

## 1   Introduction

In this paper, we consider the nonsmooth convex optimization (NSCO) problem with the First-order Oracle (FO) and the Projection Oracle (PO) defined as:

$$\text{NSCO}: \min_{x} \ f(x), \ \text{s.t. } x \in \mathcal{X} \ , \ \ \text{FO}(x) \in \partial f(x), \ \text{and PO}(x) = \mathcal{P}_{\mathcal{X}}(x) = \operatorname*{argmin}_{y \in \mathcal{X}} \|y - x\|_2^2, \ (1)$$

where $f : \mathbb{R}^d \to \mathbb{R}$ is a convex Lipschitz-continuous function, and $\mathcal{X} \subseteq \mathbb{R}^d$ is a convex constraint. When queried at a point $x$, FO returns a subgradient of $f$ at $x$ and PO returns the projection of $x$ onto $\mathcal{X}$. NSCO is a fundamental problem with a long history and several important applications including support vector machines (SVM) [12], robust learning [44], and utility maximization in finance [82].

Finding an $\varepsilon$-suboptimal solution for this problem requires $\Omega(\varepsilon^{-2})$ FO calls in the worst case, when the dimension $d$ is large [64]. This lower bound is tightly matched by the projected subgradient method (PGD). Unfortunately, PGD also uses one PO call after every FO call, resulting in a PO calls

| | Randomized Smoothing dimension dependent | State-of-the-art dimension-free | Our results (Theorems 1 and 2) | Lower bound |
|---|---|---|---|---|
| SFO | $\mathcal{O}((G^2 + \sigma^2)/\varepsilon^2)$ [27] | $\mathcal{O}((G^2 + \sigma^2)/\varepsilon^2)$ [65] | $\mathcal{O}((G^2 + \sigma^2)/\varepsilon^2)$ | $\Omega((G^2 + \sigma^2)/\varepsilon^2)$ [64] |
| PO | $\mathcal{O}(d^{1/4}G/\varepsilon)$ [27] | $\mathcal{O}(G^2/\varepsilon^2)^\star$ [65] | $\mathcal{O}(G/\varepsilon)$ | Open problem |
| SFO | $\mathcal{O}(\sqrt{d}\,(G^2 + \sigma^2)^2/\varepsilon^4)$ [54] | $\mathcal{O}((G^2 + \sigma^2)/\varepsilon^2)^\dagger$ | $\mathcal{O}((G^2 + \sigma^2)/\varepsilon^2)$ | $\Omega((G^2 + \sigma^2)/\varepsilon^2)$ [64] |
| LMO | $\mathcal{O}(\sqrt{d}\,G^2/\varepsilon^2)$ [54]* | $\mathcal{O}((G^2 + \sigma^2)^2/\varepsilon^4)^\dagger$ | $\mathcal{O}(G^2/\varepsilon^2)$ | $\Omega(G^2/\varepsilon^2)$ [54] |

Table 1: Comparison of SFO (3), PO (1) & LMO (2) calls complexities of our methods and state-of-the-art algorithms, and corresponding lower-bounds for finding an approximate minimizer of a $d$-dimensional NSCO problem (1). We assume that $f$ is convex and $G$-Lipschitz continuous, and is accessed through a stochastic subgradient oracle with a variance of $\sigma^2$. *requires using a minibatch of appropriate size, $\dagger$approximates projections of PGD with FW method (FW-PGD, see Appendix B.2).

complexity (PO-CC)—the number of times PO needs to be invoked—of $\Theta(\varepsilon^{-2})$. This can be a major bottleneck in solving several practical problems like collaborative filtering [79], where the cost of a PO is often higher than the cost of an FO call. This begs the natural question, which surprisingly is largely unexplored in the general nonsmooth optimization setting: *Can we design an algorithm whose PO calls complexity is significantly better than the optimal FO calls complexity $O(\varepsilon^{-2})$?*

In this work, we answer the above question in the affirmative. Our first key contribution is MOreau Projection Efficient Subgradient method (MOPES), that obtains an $\varepsilon$-suboptimal solution using only $\mathcal{O}(\varepsilon^{-1})$ PO calls, while still ensuring that the FO calls complexity (FO-CC)—the number of times FO needs to be invoked—is optimal, i.e., $\mathcal{O}(\varepsilon^{-2})$. This requires a mild assumption that the function $f$ extends to a slightly larger neighborhood of the constraint set $\mathcal{X}$. Concretely, we assume that $f$ is Lipschitz continuous in this neighborhood and FO can be queried at points in this neighborhood. To the best of our knowledge, our result is the first improvement over the $\mathcal{O}(\varepsilon^{-2})$ PO calls of PGD for minimizing a general nonsmooth Lipschitz continuous convex function.

We achieve this by carefully combining Moreau-Yosida regularization with accelerated first-order methods [62, 81]. As accelerated methods cannot be directly applied to a nonsmooth $f$, we can instead apply them to minimize its Moreau envelope, which is smooth (as long as $f$ is Lipschitz continuous). Although this idea has been explored, for example, in [25, 9], PO-CC has remained $\mathcal{O}(\varepsilon^{-2})$, unless a much stronger and unrealistic oracle is assumed [9] with a direct access to the gradient of Moreau envelope. The key idea in breaking this barrier is to separate out the dependence on FO calls of $f$ from PO calls to $\mathcal{X}$ by: $(a)$ using Moreau-Yosida regularization to *split* the original problem into a composite problem, where one component consists of an unconstrained optimization of the function $f$ and the other consists of a simple constrained optimization over the set $\mathcal{X}$; and $(b)$ applying the gradient sliding algorithm [55] on this joint problem to ensure the above mentioned bounds for both FO and PO calls. We note that our results are limited to the Euclidean norm, since our results crucially depend on smoothness of the Moreau envelope and its regularizer, which is not known for Moreau envelopes based on general Bregman divergences [7].

In some high-dimensional problems, even a single call to the PO can be computationally prohibitive. A popular alternative, pioneered by Frank and Wolfe [28], is to replace PO by a more efficient Linear Minimization Oracle (LMO), which returns a minimizer of any linear functional $\langle g, \cdot \rangle$ over the set $\mathcal{X}$.

$$\text{LMO}\,(g) \in \operatorname*{argmin}_{s \in \mathcal{X}} \langle g, s \rangle \qquad (2)$$

Linear minimization is much faster than projection in several practical ML applications such as a nuclear norm ball constrained problems [15], video-narration alignment [1], structured SVM [51], and multiple sequence alignment and motif discovery [89]. LMO based methods have an important additional benefit of producing solutions that preserve desired structures such as sparsity and low rank. For *smooth* $f$, there is a long history of conditional gradient (Frank-Wolfe) methods that use $\mathcal{O}(\varepsilon^{-1})$ LMO calls and $\mathcal{O}(\varepsilon^{-1})$ FO calls to achieve $\varepsilon$-suboptimality, which achieve optimal LMO-CC [45]. For *nonsmooth* functions, starting from the work of [84], several approaches have been proposed, some under more assumptions. The best known upper bound on LMO calls is $\mathcal{O}(\sqrt{d}\varepsilon^{-2})$ which is achieved at the expense of significantly larger $\mathcal{O}(\varepsilon^{-4})$ FO calls. Details of these are in Section 1.1.

Our second key contribution is the algorithm MOLES, which obtains an $\varepsilon$-suboptimal solution using the optimal $\mathcal{O}(\varepsilon^{-2})$ LMO and FO calls, without any additional dimension dependence. We

achieve this result by extending MOPES to work with approximate projections and using the classical Frank-Wolfe (FW) method [28] to implement these approximate projections using LMO calls.

Finally, both of our methods extend naturally to the Stochastic First-order Oracle (SFO) setting, where we have access only to stochastic versions of the function's subgradients. Stochastic versions of MOPES and MOLES still achieve the the same PO/LMO calls complexities as deterministic counterparts, while the SFO calls complexity (SFO-CC) is $\mathcal{O}\left((1 + \sigma^2)\epsilon^{-2}\right)$, where $\sigma^2$ is the variance in SFO. This again matches information theoretic lower bounds [64].

**Contributions**: We summarize our contributions below and in Table 1. We assume that the function $f$ extends to a slightly larger neighborhood of the constraint set $\mathcal{X}$ i.e., $f$ continues to be Lipschitz continuous and (S)FO can be queried in this neighborhood.

- We introduce MOPES and show that it is guaranteed to find an $\varepsilon$-suboptimal solution for any constrained nonsmooth convex optimization problem using $\mathcal{O}(\varepsilon^{-1})$ PO calls and optimal $\mathcal{O}(\varepsilon^{-2})$ SFO calls. To the best of our knowledge, for the general problem, this achieves the first improvement over $\mathcal{O}(\varepsilon^{-2})$ PO-CC and SFO-CC of stochastic projected subgradient method (PGD).
- For LMO setting, we extend our method to design MOLES, that achieves the optimal SFO-CC and LMO-CC of $\mathcal{O}(\varepsilon^{-2})$, and improves over the best known LMO-CC by $\sqrt{d}$.
- We also empirically evaluate MOPES and MOLES on the popular nuclear norm constrained Matrix SVM problem [85], where they achieve significant speedups over their corresponding baselines.
- Our main technical novelty is the use Moreau-Yosida regularization to separate out the constraint (PO/LMO) and function (SFO) accesses into two parts of a composite optimization problem. This enables a better control of how many times each of these oracles are accessed. This idea might be of independent interest, whenever a trade-off between PO-CC/LMO-CC and SFO-CC is desirable.

## 1.1 Related Work

**Nonsmooth convex optimization**: Nonsmooth convex optimization has been the focal point of several research works for past few decades. [64] provided information theoretic lower bound of FO calls $O(\varepsilon^{-2})$ to obtain $\varepsilon$-suboptimal solution, for the general problem. This bound is matched by the PGD method introduced independently by [34] and [59], which also implies a PO-CC of $O(\varepsilon^{-2})$. Recently, several faster PGD style methods [50, 78, 87, 48] have been proposed that exploit more structure in the given optimization function, e.g., when the function is a sum of a smooth and a nonsmooth function for which a *proximal* operator is available [8]. But, to the best of our knowledge, such works do not explicitly address PO-CC and are mainly concerned about optimizing FO-CC. Thus, for the worst case nonsmooth functions, these methods still suffer from $O(\varepsilon^{-2})$ PO-CC.

**Smoothed surrogates**: Smoothing of the nonsmooth function is another common approach in solving them [62, 66]. In particular, randomized smoothing [27, 9] techniques have been successful in bringing down FO-CC w.r.t. $\varepsilon$ but such improvements come at the cost of dimension factors. For example, [27, Corollary 2.4] provides a randomized smoothing method that has $O(d^{1/4}/\varepsilon)$ PO-CC and $O(\varepsilon^{-2})$ FO-CC. Our MOPES method guarantees significantly better PO-CC than PGD that is still *independent* of dimension.

**One or $\log(1/\epsilon)$ projection methods**: Starting with the work of [61], several recent works [91, 17, 88] have proposed methods that require only *one* or $\log(1/\epsilon)$ projections, under a variety of conditions on the optimization function like smoothness and strong convexity. However, these methods require that the constraint set can be written as $c(x) \leq 0$ and they require access to $\nabla c(x)$—the gradient of $c$–in *each* iteration. Hence, for the general nonsmooth functions, they will require at least $O(\varepsilon^{-2})$ accesses to gradients of the set's functional form. On the other hand, our method is required to access the set at only $O(\varepsilon^{-1})$ points. Furthermore, for several practical problems, the computational complexities of computing $\nabla c(x)$ and projecting are similar. For example, when $c(x) = \|x\|_{\mathrm{nuc}} - r$ where $\| \cdot \|_{\mathrm{nuc}}$ denotes the nuclear norm (see Section 4), then both gradient of $c(x)$ as well as PO requires computation of a full-SVD of $x$.

**Frank-Wolfe methods:** FW or *conditional gradient* method [28, 59] for smooth convex optimization, which uses LMO, has found renewed interest in machine learning [92, 45] due to the efficiency of computing LMO over PO [33], and its ability to ensure atomic structure and provide coreset guarantees [22]. Over the last decade, several variants of FW method and their analyses have been proposed [54, 29, 3, 31, 58, 68, 14], and FW has been extended to stochastic nonconvex

[49, 39, 75, 76, 5, 37] and online [38, 30, 52, 18, 86, 40] settings. However these methods provide dimension-free LMO-CC and SFO-CC only for smooth functions, and further it is known that FW fails to converge if subgradients are used instead of gradients [68].

**Nonsmooth Frank-Wolfe methods:** [84] posed an interesting question in the domain of nonsmooth optimization with LMO: can LMO-CC be reduced from the $\mathcal{O}(\varepsilon^{-4})$ bound (achieved by PGD with PO implemented via LMO: FW-PGD, see Appendix B.2) without increasing FO-CC significantly. On the lower bound side, [54] showed that $\mathcal{O}(\varepsilon^{-2})$ LMO calls are necessary. On algorithmic side, several randomized smoothing approaches combined with Frank-Wolfe methods were proposed, and can reduce LMO-CC to $\mathcal{O}(d^{1/2}\varepsilon^{-2})$. But, they come at the expense of increased $\mathcal{O}(d^{1/2}\varepsilon^{-4})$ FO calls [54, improving Theorem 5][1]. If we allow stronger oracles or additional structure in the problem, the complexity can be significantly improved. Assuming a stronger than LMO oracle introduced in [84], [73] shows that $\mathcal{O}(1/\varepsilon^2)$ LMO-CC and FO-CC are achievable for a special class of problems with low curvatures. Another popular setting is when the nonsmooth problem admits a *smooth* convex-concave saddle point reformulation [35, 23, 72, 36, 41, 42, 32, 60]. Among these the best complexity is achieved by semi-proximal mirror-prox [41] which uses $\mathcal{O}(\varepsilon^{-2})$ LMO and $\mathcal{O}(\varepsilon^{-1})$ FO calls. However, for the general nonsmooth convex optimization problem with LMO, the problem posed by [84] remained open, and is resolved by our MOLES method that achieves the optimal $\mathcal{O}(\varepsilon^{-2})$ LMO-CC and FO-CC.

## 2  Preliminaries and Notations

We consider Nonsmooth Convex Optimization with FO and PO (1) or LMO (2) accesses. Let $\mathcal{X} \subset \mathbb{R}^d$ be a closed convex set of diameter $D_\mathcal{X} := \max_{x_1, x_1 \in \mathcal{X}} \|x_1 - x_2\|$, where $\|\cdot\|$ is the Euclidean norm which corresponds to the inner product $\langle\cdot, \cdot\rangle$. Let $\mathcal{X}$ be enclosed in a closed convex set $\mathcal{X}' \subseteq \mathbb{R}^d$ to which it is easy to project, i.e. $\mathcal{X} \subset \mathcal{X}'$. For simplicity, let $\mathcal{X}'$ be a Euclidean ball of radius $R \leq D_\mathcal{X}$ around origin. We can satisfy $R = D_\mathcal{X}$ by re-centering $\mathbb{R}^d$ around any feasible point of $\mathcal{X}$. We assume $f : \mathcal{X}' \to \mathbb{R}$ to be a proper, lower semi-continuous (l.s.c.), convex Lipschitz function. We use $\partial f(x)$ to denote sub-differential of $f$ at $x$, and if $f$ is differentiable we use $\nabla f(x)$ to denote its gradient at $x$. We assume a first-order oracle (FO) can provide access to some subgradient at any point in $\mathcal{X}'$, i.e. $FO(x) \in \partial f(x)$.

**Definition 1.** *A function $f : \mathcal{X}' \to \mathbb{R}$ is G-Lipschitz if and only if, $|f(y) - f(x)| \leq G\|y - x\|$ for all $x, y \in \mathcal{X}'$. For a convex $f$, this is equivalent to: $\max_{x \in \mathcal{X}'} \max_{g \in \partial f(x)} \|g\| \leq G$.*

**Definition 2.** *A function $f : \mathcal{X}' \to \mathbb{R}$ is $\mu$-strongly convex if and only if, $\frac{\mu}{2}\|y - x\|^2 + \langle g, y - x\rangle + f(x) \leq f(y)$, for all $x, y \in \mathcal{X}'$ and $g \in \partial f(x)$. Similarly, a differentiable function $f : \mathcal{X}' \to \mathbb{R}$ is said to be L-smooth if and only if, $f(y) \leq f(x) + \langle \nabla f(x), y - x\rangle + \frac{L}{2}\|y - x\|^2$ for all $x, y \in \mathcal{X}'$.*

In addition to FO, we also consider problems with stochastic FO (SFO) access, which computes stochastic subgradient of a point $x$ with variance $\sigma^2$, as defined below:

$$SFO(x) := \widehat{g}, \text{ where } \mathbb{E}[\widehat{g} \,|\, x] = g \text{ for some } g \in \partial f(x), \text{ and } \mathbb{E}[\|\widehat{g} - g\|^2 \,|\, x] \leq \sigma^2. \quad (3)$$

**Moreau Envelope:** The key idea behind our method is to use "smoothed" version of the function via its Moreau envelope [63, 90] defined below.

**Definition 3.** *For a proper l.s.c. convex function $f : \mathcal{X}' \to \mathbb{R} \cup \{\infty\}$ defined on a closed convex set $\mathcal{X}'$ and $\lambda > 0$, its* Moreau-(Yosida) envelope *function, $f_\lambda : \mathcal{X}' \to \mathbb{R}$, is given by*

$$f_\lambda(x) = \min_{x' \in \mathcal{X}'} f(x') + \frac{1}{2\lambda}\|x - x'\|^2, \quad \text{for all } x \in \mathcal{X}'. \quad (4)$$

*Furthermore, the* prox *operator is defined:* $\text{prox}_{\lambda f}(x) := \text{argmin}_{x' \in \mathcal{X}'} f(x') + \frac{1}{2\lambda}\|x - x'\|^2.$

When $f$ is clear from context, we will use $\hat{x}_\lambda(x)$ to denote $\text{prox}_{\lambda f}(x)$. Note that this definition of Moreau envelope is not standard as $x'$ is constrained to $\mathcal{X}' \subseteq \mathbb{R}^d$. However, the following lemma (whose proof is in Appendix C.3) shows that this Moreau envelope and the prox operator still satisfies most useful properties of the standard definition.

**Lemma 1.** *For a closed convex set $\mathcal{X}'$, a convex proper l.s.c. function $f : \mathcal{X}' \to \mathbb{R} \cup \{\infty\}$ and $\lambda > 0$, the following hold for any $x \in \mathcal{X}'$.*
*(a) $\hat{x}_\lambda(x)$ is unique and $f(\hat{x}_\lambda(x)) \leq f_\lambda(x) \leq f(x)$,*
*(b) $f_\lambda$ is convex, differentiable, $1/\lambda$-smooth and $\nabla f_\lambda(x) = (1/\lambda)(x - \hat{x}_\lambda(x))$, and,*
*(c) if $f$ is $G$-Lipschitz continuous, then, $\|\hat{x}_\lambda(x) - x\| \leq G\lambda$, and $f(x) \leq f_\lambda(x) + G^2\lambda/2$.*

This lemma implies that, to find an $\varepsilon$-approximate minima of a nonsmooth $f$, one can instead minimize $f_\lambda$ and achieve a faster convergence by exploiting its smoothness. Concretely, if $f$ is $G$-Lipschitz and $\lambda = O(\varepsilon/G^2)$, and Lemma 1(c) ensures that solving $f_\lambda$ up to $O(\varepsilon)$ accuracy guarantees $O(\varepsilon)$ accuracy in the original minimization of $f$ (Lemma 2). This insight allows us to design a simple method that can reduce PO-CC but at the cost of a higher FO-CC. Next section starts with this result as a warm-up and then presents our method, which ensures reduced PO-CC with optimal FO-CC.

## 3 Main Results

We present our main results in this section. We first present the main ideas in Section 3.1 and then the results for PO and LMO settings in Sections 3.2 and 3.3 respectively.

### 3.1 Main Ideas

We are interested in the NSCO problem (1). As discussed in the previous section, instead of optimizing $f(x)$ over $\mathcal{X}$, we can instead optimize the Moreau envelope function $f_\lambda(x)$ with $\lambda = O(\epsilon)$ to get $\epsilon$-suboptimality. Since by Lemma 1, $f_\lambda(\cdot)$ is a $1/\lambda$-smooth convex function, a straightforward approach is to iteratively optimize $f_\lambda(x)$ using Nesterov's accelerated gradient descent (AGD) [69] method. But to get gradients of $f_\lambda(x)$, we will need to solve the *inner problem* (4) approximately.

A key insight is that since the inner problem does not involve the constraint set $\mathcal{X}$, PO calls are not required in inner steps for estimating $\nabla f_\lambda(x)$. So the total number of PO calls required is equal to the total number of outer steps in minimizing $f_\lambda(x)$, which for Nesterov's AGD is $\mathcal{O}(1/\sqrt{\lambda\varepsilon}) = \mathcal{O}(\varepsilon^{-1})$. We see that this already improves over the $\mathcal{O}(\varepsilon^{-2})$ projections of PGD. However, since $\nabla f_\lambda(x)$ needs to be estimated to a good accuracy, the total number of FO calls, including in the inner loop, turns out to be $\mathcal{O}(\varepsilon^{-3})$, which is worse than the optimal $\mathcal{O}(\varepsilon^{-2})$ FO calls of PGD.

Similarly, when we have access to LMO for $\mathcal{X}$, we could optimize $f_\lambda$ using FW [28, 45], with total number of outer steps $= \mathcal{O}(1/\lambda\varepsilon) = \mathcal{O}(\varepsilon^{-2})$, and hence the total number of LMO calls is $\mathcal{O}(\varepsilon^{-2})$. However, this again leads to suboptimal $\mathcal{O}(\varepsilon^{-4})$ FO calls. We can improve the FO-CC to $\mathcal{O}(\varepsilon^{-3})$ by using the conditional gradient sliding algorithm [56] instead of FW method, but this is still worse than the optimal $\mathcal{O}(\varepsilon^{-2})$ FO calls.

In order to achieve optimal number of FO calls, we directly optimize the Moreau envelope through the following joint optimization.

$$\min_{x \in \mathcal{X}, x' \in \mathcal{X}'} \left[ \Psi_\lambda(x, x') := f(x') + \psi_\lambda(x, x') \right] \quad \text{where} \quad \psi_\lambda(x, x') := \frac{1}{2\lambda}\|x' - x\|^2, \quad (5)$$

where the function $\Psi_\lambda : \mathcal{X}' \times \mathcal{X}' \to \mathbb{R}$ is convex in the joint variable $(x, x')$. The main advantage of this new form is that, this is a composite optimization problem with a nonsmooth part (corresponding to $f(x')$) and a $2/\lambda$-smooth part (corresponding to $(1/2\lambda)\|x' - x\|^2$) with the constrained variable $x \in \mathcal{X}$ only appearing in the smooth part. Now, by the following lemma, an approximate minimizer of $\Psi_\lambda$, is also an approximate minimizer of the Moreau envelope $f_\lambda$, and further if $\lambda = \varepsilon/G^2$, it is also an approximate minimizer of the original function $f$. A proof is provided in Appendix C.1.

**Lemma 2.** *Under the same assumptions as in Lemma 1, let $\mathcal{X} \subseteq \mathcal{X}'$ be a convex subset and $\Psi_\lambda$ be defined as in (5). Then, $(i)$ $\min_{x \in \mathcal{X}} \min_{x \in \mathcal{X}'} \Psi_\lambda(x, x') = \min_{x \in \mathcal{X}} f_\lambda(x) \leq \min_{x \in \mathcal{X}} f(x)$, and $(ii)$ for any random vectors $(x_\varepsilon, x'_\varepsilon) \in \mathcal{X} \times \mathcal{X}'$, $\mathbb{E}[f(x_\varepsilon)] - G^2\lambda/2 \leq \mathbb{E}[f_\lambda(x_\varepsilon)] \leq \mathbb{E}[\Psi_\lambda(x_\varepsilon, x'_\varepsilon)]$.*

Our algorithm essentially solves (5) using Gradient Sliding [55] and Conditional Gradient Sliding [56] frameworks, which are optimal for minimizing composite problems of the form (5) for the PO and LMO settings respectively. The resulting algorithm for PO setting, called MOPES is given in Algorithm 1. The algorithm for LMO setting, called MOLES is presented in Algorithm 2. The only difference between MOPES and MOLES is that MOLES uses FW to compute approximate projections while MOPES uses exact projections. Finally, our algorithms extend straightforwardly

---
**Algorithm 1:** MOPES: MOreau Projection Efficient Subgradient method
---

**Input:** $f$, $\mathcal{X}$, $\mathcal{X}'$, $G$, $D_{\mathcal{X}}$, $R$, $x_0$, $K$, $\tilde{D}$, $c'$, $\lambda$,

1.1  Set $x_0' = z_0' = x_0 = z_0 = x_0$

1.2  **for** $k = 1, \ldots, K$ **do**

1.3  $\quad$ Set $\beta_k = \frac{4}{\lambda k}$ , $\gamma_k = \frac{2}{k+1}$ , and $T_k = \left\lceil \frac{(4G^2+\sigma^2)\lambda^2 K k^2}{2\tilde{D}} \right\rceil$

1.4  $\quad$ Set $(y_k, y_k') = (1 - \gamma_k) \cdot (x_{k-1}, x_{k-1}') + \gamma_k \cdot (z_{k-1}, z_{k-1}')$

1.5  $\quad$ Set $z_k = \mathcal{P}_{\mathcal{X}}\left(z_{k-1} - \frac{1}{\beta_k} \cdot \nabla_{y_k} \Psi_\lambda(y_k, y_k')\right)$ (1) $\qquad$ // Note $\nabla_{y_k}\Psi_\lambda(y_k, y_k') = \frac{y_k - y_k'}{\lambda}$

1.6  $\quad$ Set $(z_k', \tilde{z}_k') = \texttt{Prox-Slide}\left(\nabla_{y_k'}\psi_\lambda(y_k, y_k'), z_{k-1}', \beta_k, T_k\right)$ $\quad$ // $\nabla_{y_k'}\psi_\lambda(y_k, y_k') = \frac{y_k' - y_k}{\lambda}$

1.7  $\quad$ Set $(x_k, x_k') = (1 - \gamma_k) \cdot (x_{k-1}, x_{k-1}') + \gamma_k \cdot (z_k, \tilde{z}_k')$

$\quad$ **Output:** $(x_K, x_K')$

1.8  $\texttt{Prox-Slide}$($g$, $u_0$, $\beta$, $T$) *// Approx. resolve* $\text{prox}_{f/\beta}(u_0' - g/\beta)$[55]**:**

1.9  $\quad$ Set $\tilde{u}_0 = u_0$

1.10 $\quad$ **for** $t = 1, \ldots, T$ **do**

1.11 $\quad\quad$ Set $\theta_t = \frac{2(t+1)}{t(t+3)}$, $\widehat{g}_{t-1} = \text{SFO}(u_{t-1})$ (3)

1.12 $\quad\quad$ Set $\widehat{u}_t = u_{t-1} - \frac{1}{(1+t/2)\beta} \cdot \left(\widehat{g}_{t-1} + \beta(u_{t-1} - (u_0 - g/\beta))\right)$

$\quad\quad\quad$ // subgradient method step for $\phi(u) := f(u) + \frac{\beta}{2}\|u - (u_0 - \frac{g}{\beta})\|^2$

1.13 $\quad\quad$ Set $u_t = \widehat{u}_t \cdot \min\left(1, R/\|u_t\|\right)$ $\qquad$ // projection of $\widehat{u}_t$ onto $\mathcal{X}'$: $\mathcal{P}_{\mathcal{X}}'(\mathbf{u}_t)$

1.14 $\quad\quad$ Set $\tilde{u}_t = (1 - \theta_t) \cdot \tilde{u}_{t-1} + \theta_t \cdot u_t$

1.15 $\quad$ **return** $(u_T, \tilde{u}_T)$

---

to the case of stochastic subgradients through a stochastic first order oracle (SFO) and the resulting bounds depend on the variance of SFO in addition to the Lipschitz constant of $f(\cdot)$.

### 3.2 MOreau Projection Efficient Subgradient (MOPES) method

A pseudocode of our algorithm MOPES is presented in Algorithm 1. At a high level, MOPES is an inexact Accelerated Proximal Gradient method (APGD) [67, 8] scheme which tries to implement Nesterov's AGD algorithm on $\Psi_\lambda(x, x')$. Now, standard AGD updates for solving $\min_{x \in \mathcal{X}, x'} \Psi_\lambda(x, x')$, *if $\Psi_\lambda$ were smooth* are:

$$
\begin{aligned}
&\beta_k \leftarrow 4/\lambda k \ , \ \gamma_k \leftarrow 2/(k+1) \\
&(y_k, y_k') \leftarrow (1 - \gamma_k)(x_{k-1}, x_{k-1}') + \gamma_k(z_{k-1}, z_{k-1}') \\
&z_k \leftarrow \mathcal{P}_{\mathcal{X}}(z_{k-1} - \nabla_{y_k}\Psi_\lambda(y_k, y_k')/\beta_k), \ \ z_k' \leftarrow z_{k-1}' - \nabla_{y_k'}\Psi_\lambda(y_k, y_k')/\beta_k, \\
&(x_k, x_k') \leftarrow (1 - \gamma_k)(x_{k-1}, x_{k-1}') + \gamma_k(z_k, z_k').
\end{aligned}
\tag{6}
$$

MOPES essentially implements the above updates, but as $\Psi_\lambda$ is nonsmooth in $x'$, we use prox steps for the $x'$ variable instead of the GD steps. The prox step—$\text{prox}_{f/\beta_k}\left(z_{k-1}' - \nabla_{y_k'}\psi_\lambda(y_k, y_k')/\beta_k\right)$— is implemented via $\texttt{Prox-Slide}$ procedure (see Line 1.6), which is the standard subgradient method applied to a strongly convex function $\phi$ (see Line 1.12). Now, $\texttt{Prox-Slide}$ procedure outputs two points $(z_k', \tilde{z}_k')$ which are the final and average iterates, respectively, of the subgradient method, This achieves optimal FO-CC by exploiting strong convexity of $\phi$. If we were to use only the average of the iterates, the FO-CC would increase by a factor of $\mathcal{O}(\varepsilon^{-1})$ (see the failed attempt in Appendix A.1).

Note that MOPES needs only a PO call & no FO call in Line 1.5, and only a FO/SFO call in Line 1.11. Therefore, we bound below, the total number of PO calls $K$ and the number of FO/SFO calls $K \cdot T$.

**Theorem 1.** *Let $f : \mathcal{X}' \to \mathbb{R}$ be a G-Lipschitz continuous proper l.s.c. convex function equipped with a SFO with variance $\sigma^2$, and $\mathcal{X} \subseteq \mathcal{X}' = B(0, R)$ be some convex subset equipped with a projection oracle $\mathcal{P}_{\mathcal{X}}$ and contained inside the Euclidean ball of radius $R$ around origin. If we run MOPES (Algorithm 1) with inputs $\lambda = \varepsilon/G^2$, $\tilde{D} = c\|x_0 - x^*\|^2$ and $K = \lceil 2\sqrt{(10 + 8c)}G\|x_0 - x^*\|/\varepsilon \rceil$ for any absolute constant $c > 0$ and $x^* \in \arg\min_{x \in \mathcal{X}} f(x)$, then, using $\mathcal{O}\left(\frac{G\|x_0 - x^*\|}{\varepsilon}\right)$ PO calls and $\mathcal{O}\left(\frac{(G^2 + \sigma^2)\|x_0 - x^*\|^2}{\varepsilon^2}\right)$ FO calls, it outputs $x_K$ satisfying $f(x_K) - \min_{x \in \mathcal{X}} f(x) \le \varepsilon$.*

**Remarks**: Note that FO-CC is same as that of PGD (up to constants) while PO-CC is significantly better. A natural open question is if PO-CC can be further reduced. Also, MOPES requires querying of SFO/FO at $u_{t-1}$ which is not necessarily in $\mathcal{X}$ but is always in $\mathcal{X}'$ (Line 1.11). Recall from Section 2 that $\mathcal{X}'$ is a Euclidean ball of radius $R \leq D_{\mathcal{X}}$ around origin. Being able to query SFO/FO in $\mathcal{X}'$ seems like a mildly stricter requirement than the standard requirement of querying on $\mathcal{X}$ only, but for most practical problems this seems feasible. Even if $f$ is unknown outside of $\mathcal{X}$, theoretically we could work with its convex extension to the entire space, which remains $G$ Lipschitz (see Section 6). Also, notice that the guarantee only depends on the diameter $D_{\mathcal{X}}$ of the constraint set $\mathcal{X}$ and not the radius $R$ of the enclosing set $\mathcal{X}'$. This is so because the first-order method only depends on the distance from initial point $(x_0, x_0')$ to the desired solution $(x^*, x^*)$, which is $\mathcal{O}(\|x_0 - x^*\|) = \mathcal{O}(D_{\mathcal{X}})$, as $x_0' = x_0$. Finally, for simplicity of exposition, we provide desired suboptimality $\epsilon$ as an input to MOPES–in practice, we can remove this assumption by using standard doubling trick [80, Algorithm 6].

See Appendices A.2 and C.2.1 for a proof sketch and a detailed proof, respectively, of Theorem1. At a high level, our proof uses a potential function [6] for analyzing APGD, combines it with Proposition 1 which provides a fast convergence guarantee on `Prox-Slide` iterates, and then apply standard APGD proof techniques [81] to obtain the final result.

**Proposition 1** (Proposition 3, informal). *For some $0 < \tau_k \leq 1.5$, output of* `Prox-Slide` *satisfies*

$$\phi_k(\widetilde{z}_k') - \phi_k(x') + \frac{\beta_k}{2}\|z_{k-1}' - x'\|^2 \leq \frac{\beta_k}{2}[\tau_k\|z_{k-1}' - x'\|^2 - \tau_{k+1}\|z_k' - x'\|^2] + \frac{16\,G^2}{\beta_k T_k}.$$

### 3.3  MOreau Linear minimization oracle Efficient Subgradient (MOLES) method

---

**Algorithm 2:** MOLES: MOreau Linear minimization oracle Efficient Subgradient method

Use the same steps as MOPES (Algorithm 1), but replace Line 1.5 with:

**2.5** Set $z_k = $ `FW-Based-Projection`$(z_{k-1} - \frac{1}{\beta_k} \cdot \nabla_{y_k}\Psi_\lambda(y_k, y_k'),\ z_{k-1},\ \lceil\frac{7KD_{\mathcal{X}}^2}{c'\tilde{D}}\rceil)$

**2.16** `FW-Based-Projection`$(z, u_0, \hat{T})$:

    // $\hat{T}$ steps of standard Frank-Wolfe for $\min_{u\in\mathcal{X}}\|u - z\|^2$

**2.17**     **for** $t = 1,\ldots,\hat{T}$ **do**

**2.18**         Set $s_t = \text{LMO}\,(u_{t-1} - z)$

**2.19**         Set $u_t = ((t-1)\cdot u_{t-1} + 2\cdot s_t)/(t+1)$

**2.20**     **return** $u_{\hat{T}}$

---

We now present our results for the LMO setting. A pseudocode of our algorithm, MOLES, is presented in Algorithm 2. MOLES does exactly the same steps as in MOPES (Algorithm 1), except that the projection in Line 1.5 of MOPES is estimated using the LMO and Frank-Wolfe algorithm. At the outer-step $k$, the output $z_k$ of `FW-Based-Projection`, which uses $\hat{T} = \mathcal{O}(1/\varepsilon)$ LMO calls to approximately project, satisfies the following bound on the projection problem's Wolfe dual gap [45]:

$$\max_{s\in\mathcal{X}} \beta_k \left\langle z_k - \left(z_{k-1} - (1/\beta_k)\cdot\nabla_{y_k}\Psi_\lambda(y_k, y_k')\right), z_k - s \right\rangle \leq 4c'\tilde{D}/\lambda K k \qquad (7)$$

In practice we can use the above condition as a stopping criterion for `FW-Based-Projection`. The following theorem, a proof of which is in Appendix C.2.2, provides the convergence guarantee.

**Theorem 2.** *Let $f : \mathcal{X}' \to \mathbb{R}$ be a $G$-Lipschitz continuous proper l.s.c. convex function equipped with an SFO with variance $\sigma^2$, and $\mathcal{X} \subseteq \mathcal{X}' = B(0, R)$ be some convex subset of diameter $D_{\mathcal{X}}$ equipped with an LMO and contained inside the Euclidean ball of radius around origin. If we run MOLES (Algorithm 2) with inputs $\lambda = \varepsilon/G^2$, $\tilde{D} = cD_{\mathcal{X}}$, $K = \lceil 2\sqrt{10 + 8c(1 + c')}G\|x_0 - x^*\|/\varepsilon\rceil$, for some absolute constants $c, c' > 0$ and $x^* \in \arg\min_{x\in\mathcal{X}} f(x)$, then, using $\mathcal{O}(\frac{G^2 D_{\mathcal{X}}^2}{\varepsilon^2})$ LMO calls and $\mathcal{O}(\frac{(G^2 + \sigma^2)D_{\mathcal{X}}^2}{\varepsilon^2})$ FO calls it outputs $x_K$ satisfying $f(x_K) - \min_{x\in\mathcal{X}} f(x) \leq \varepsilon$.*

**Remarks**: Thus our algorithm obtains the optimal $\mathcal{O}(\varepsilon^{-2})$ dimension independent FO-CC and LMO-CC for general nonsmooth functions [54]. Similar to MOPES, here also, we require FO/SFO of $f$ to be well-defined in $\mathcal{X}'$. If $f$ is a maximum of smooth convex functions, then we can get similar PO-CC

by applying min-max saddle point approaches [41]. But even for such functions, it is non-trivial to extend saddle point approaches to stochastic FO, which is important in practice. In contrast, our result matches the optimal FO-CC (on all key parameters) of unconstrained stochastic-PGD method.

## 4   Applications

We first explain the gain of MOPES in practical applications. One of the main applications of our method is Empirical Risk Minimization (ERM) with nonsmooth loss functions. For a nonsmooth loss $f_i$ for the $i$th training example in a set of $n$ examples, the general form of ERM is:

$$\min_{x \in \mathcal{X} \subseteq \mathbb{R}^d} \frac{1}{n} \sum_{i=1}^{n} f_i(x) \quad \text{For example,} \quad \min_{X \in \mathbb{R}^{m \times p}; \|X\|_{\text{nuc}} \leq r} \frac{1}{n} \sum_{i=1}^{n} \max(0, 1 - b_i \langle X, A_i \rangle), \quad (8)$$

which is known as the *low rank SVM* [85, 83, 74] as the nuclear norm constraint induces low rank solutions. As the cost of a single PO call involves a full SVD on a potentially full rank $X$, MOPES significantly improves over the competing baseline as we showcase in Fig. 1. There are numerous examples of ERMs with costly POs to a nuclear norm ball (e.g. max-margin collaborative filtering [79]), to an $\ell_1$ norm ball (e.g. sparse SVM [13, 93, 4]), and to a large number of linear constraints (e.g. robust classification [10]). One notable example is *SVM with hard constraints* on a subset of the training data, so that some predictions are constrained to be always accurate [70] (See Appendix E.2).

In all these examples, PO calls can be more costly than FO calls, making MOPES attractive. In comparison, popular *accelerated proximal point methods*, such as FISTA [8], cannot handle general nonsmooth losses. The standard *projected subgradient methods* suffer from $\mathcal{O}(\varepsilon^{-2})$ PO-CC. *Mirror descent* [64] may give better $d$ dependence, but it too requires $O(\varepsilon^{-2})$ (proximal) operations.

Now, several nonsmooth loss functions have a special structure where they can be written as a *smooth minimax problem*. Such (stochastic) problems can be solved using $\mathcal{O}_\varepsilon(\varepsilon^{-1})$ (S)FO and PO calls [66]. However, the resulting complexity scales up with the dimension $d$ or the number of samples $n$. Thus the PO-CC of the minimax formulations becomes inefficient (even with variance reduction [71, 16]), whenever $n$ or $d$ gets large. In the deterministic setting, each step of the optimization problem requires gradient of the entire empirical risk function, so for problems with large $n$ and small $\varepsilon$, total time complexity can be significantly higher than MOPES. See Appendix E.1 for exact complexities.

Further, beyond ERM, nice minimax representations might not always exist. For example, in reinforcement learning/optimal control setting, $f$ could be an (already trained) *input-convex neural network* [2, 19] approximating the Q-function over a continuous constrained action space [20].

For several of the above examples, LMO might be preferred if it is significantly more efficient than a PO call e.g., for high-dimensional low rank SVM, a LMO call only requires computing top singular vector, as opposed to full SVD required by a PO. Further, LMO-based methods have an additional benefit of preserving the desired structure of the solution, such as sparse and low rank structures [22]. This makes MOLES particularly attractive, for example, in differentially private collaborative filtering [46], where structured updates lead to improved privacy guarantees. In Appendix E, we present the details of some these examples, and give analytical comparisons to competing methods.

## 5   Empirical Results

We experimentally evaluate[2] MOPES (Algorithm 1) and MOLES (Algorithm 2) methods on a low rank SVM problem [85] of the form (8) on a subset of the Imagewoof 2.0 dataset [43]. The training data contains $n = 400$ samples $\{(A_i, y_i)\}_{i=1}^{n}$ where $A_i$ is a $224 \times 224$ grayscale image labeled using $y_i \in \{0, 1\}$. Note that the effective dimension is $d = 50176$. We use $r = 0.1$ as nuclear norm ball radius of $\mathcal{X}$. First, we compare the PO and FO efficiencies of MOPES with those of PGD with a fixed and PGD with a diminishing stepsize. In Figure 1 we plot the mean (over 10 runs) sub-optimality gap: $f(x_k) - \hat{f}^*$, of the iterates against the number of PO (top) and FO (bottom) calls, respectively, used to obtain that iterate. Next, we compare the LMO and FO efficiencies of MOLES with those of FW-PGD (see Algorithm 3 in Appendix B.2) and Randomized Frank-Wolfe (RandFW) [54, Theorem 5] methods with a fixed and diminishing stepsizes. In Figure 2 we plot the

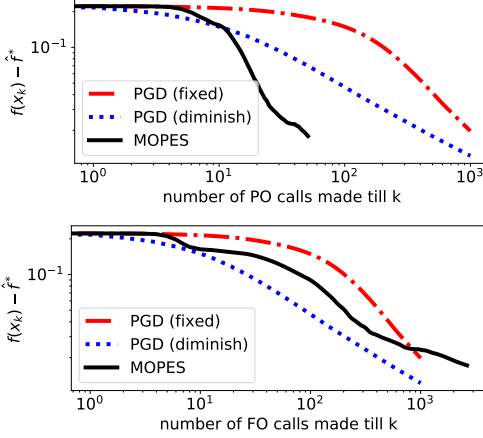
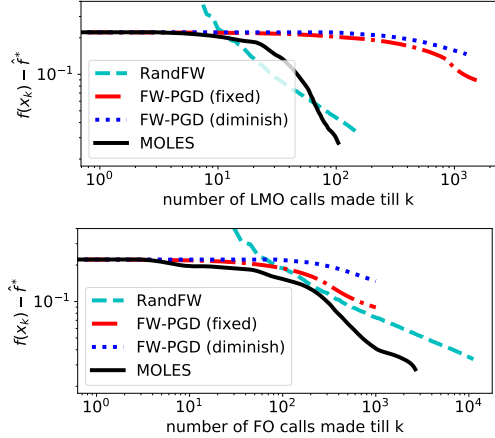

Figure 1: MOPES uses significantly fewer PO calls and comparable number of FO calls than PGD

Figure 2: MOLES uses fewer LMO calls and similar number of FO calls than FW-PGD and RandFW

mean (over 10 runs) sub-optimality gap: $f(x_k) - \hat{f}^*$, of the iterates against the number of LMO (top) and FO (bottom) calls, respectively, used to obtain that iterate. In both these plots, while MOPES/MOLES and baselines have comparable FO-CC, MOPES/MOLES is significantly more efficient in the number of PO/LMO calls, matching our Theorems 1 and 2. As the nuclear norm ball has a non-trivial projection/LMO, PO-CC/LMO-CC will dominate the total run-time as $m$ becomes larger for $X \in \mathbb{R}^{m \times m}$. Note that matrix mirror descent [47] would also require $O(\varepsilon^{-2})$ SVD based proximal operations. We provide additional experimental details in Appendix D.

## 6 Conclusion

We study a canonical problem in optimization: minimizing a nonsmooth Lipschitz continuous convex function over a convex constraint set. We assume that the function is accessed with a first-order oracle (FO) and the set is accessed with either a projection oracle (PO) or a linear minimization oracle (LMO). In this general setting, we address the fundamental question of reducing the number of accesses to the function and the set. When using projections, we introduce MOPES, and show that it finds an $\varepsilon$-suboptimal solution with $\mathcal{O}(\varepsilon^{-2})$ FO calls and $\mathcal{O}(\varepsilon^{-1})$ PO calls. This is optimal in the number of FO calls and significantly improves over competing methods in the number of PO calls (see Table 1). When using linear minimizations, we introduce MOLES, and show that it finds an $\varepsilon$-suboptimal solution with $\mathcal{O}(\varepsilon^{-2})$ FO and LMO calls. This is optimal in both the number of PO and the number of LMO calls. This resolves a question left open since [84] on designing the optimal Frank-Wolfe type algorithm for nonsmooth functions.

The two properties we need of the superset $\mathcal{X}' \supseteq \mathcal{X}$ are that (a) it is easy to project onto $\mathcal{X}'$ and (b) $f$ is $G$-Lipschitz on $\mathcal{X}'$. In our paper, we choose $\mathcal{X}'$ to be a Euclidean ball (which is easy to project to) but any other choice of $\mathcal{X}'$ which satisfies the above properties works just as well. For example, if $f$ is Lipschitz everywhere, we can set $\mathcal{X}' = \mathbb{R}^d$ and ignore the explicit projection to $\mathcal{X}'$ in line 1.13 of Algorithm 1. However, even if $f$ is $G$-Lipschitz inside the constraint $\mathcal{X}$, $f$ could (i) have unbounded Lipschitz constant, or (ii) be undefined just outside of $\mathcal{X}$. Thus an $\mathcal{X}'$ satisfying our requirements may not exist. In our experiments, we do not explicitly project onto $\mathcal{X}'$ (line 1.13) but still observed that $\|x_k - x_k'\| = \mathcal{O}(G\lambda) = \mathcal{O}(\varepsilon)$ and small, which hints that we may only need Lipschitzness over a much smaller set, say $\mathcal{X} + B(0, \mathcal{O}(G\lambda))$. Theoretically, we can work around the above issues by minimizing the convex extension $f_{\mathcal{X}} : \mathbb{R}^d \to \mathbb{R}$ of the function $f$ from the set $\mathcal{X}$, defined as $f_{\mathcal{X}}(x') := \max_{x \in \mathcal{X}} \max_{g \in \partial f(x)} f(x) + \langle g, x' - x \rangle$. The extension $f_{\mathcal{X}}$ has the same value as $f$ inside $\mathcal{X}$ and is $G$-Lipschitz everywhere. Therefore the minimization problems $\min_{x \in \mathcal{X}} f(x)$ and $\min_{x \in \mathcal{X}} f_{\mathcal{X}}(x')$ are equivalent. However, it is not clear if we can estimate the gradients of $f_{\mathcal{X}}$ efficiently. We did not find any relevant prior work and leave this question for future work.

Another possible direction of future work is developing $\varepsilon$-horizon oblivious algorithms, where we need not fix $K$ and $\varepsilon$ a priori. In our experiments, we observed that varying $\lambda$ according to $\lambda_k = \mathcal{O}(\frac{D_{\mathcal{X}}}{Gk})$ and $\beta_k = \frac{4}{\lambda_k k}$ works just as well as fixing it.

## Broader Impact

As this is foundational research that is theoretical in nature, it is hard to predict any foreseeable societal consequence.

## Funding Disclosure

Funding in direct support of this work: NSF grant 1927712, NSF grant 1929955, NSF grant 2019844, and Google faculty research award.

## Footnotes

[1] Needs tightening of [54, Theorem 5], by reducing the number of SFO calls per step by a factor of $d^{-1/2}$, i.e. $T_k = \lceil k d^{-1/2}\rceil$

[2]Code for the experiments is available at `https://github.com/tkkiran/MoreauSmoothing`

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
