[Supplementary Material]

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

# Appendix

## A Supplementary results

### A.1 Intuition behind the design of MOPES and a failed attempt

In this section we study the main ideas behind the design of MOPES method through a failed attempt. Only for the section, for simplicity, we assume that $\mathcal{X}'$ is the whole vector space, and $f$ is $G$ Lipschitz in $\mathcal{X}'$. Recall that we want to solve the problem (5),

$$\min_{x \in \mathcal{X}, x' \in \mathcal{X}'} [\Psi_\lambda(x, x') = \psi_\lambda(x, x') + f(x')]. \tag{9}$$

Notice that this is a composite objective which is a sum of a $2/\lambda$-smooth function $\psi_\lambda$ and a nonsmooth function $f$. This implies that, if we have access to the proximal operator (recall Definition 3) for $f$

$$\text{prox}_{f/t}(z) := \arg\min_{x \in \mathcal{X}'} f(x) + \frac{t}{2}\|x - z\|^2, \tag{10}$$

then theoretically we can solve this problem using accelerated proximal gradient algorithm (APGD) [8, 67, 81], which has the following update rule

$$
\begin{aligned}
&\beta_k \leftarrow 4/\lambda k \;,\; \gamma_k \leftarrow 2/(k+1) \\
&(y_k, y_k') \leftarrow (1 - \gamma_k)(x_{k-1}, x_{k-1}') + \gamma_k(z_{k-1}, z_{k-1}') \\
&z_k \leftarrow \mathcal{P}_{\mathcal{X}}(z_{k-1} - \nabla_x \psi_\lambda(y_k, y_k')/\beta_k) \\
&z_k' \leftarrow \text{prox}_{f/\beta_k}(z_{k-1}' - \nabla_{x'} \psi_\lambda(y_k, y_k')/\beta_k) \\
&(x_k, x_k') \leftarrow (1 - \gamma_k)(x_{k-1}, x_{k-1}') + \gamma_k(z_k, z_k')
\end{aligned}
\qquad , \tag{APGD}
$$

for some stepsize $1/\beta_k$ and iterate weight $\gamma_k$.

Notice that this update rule is different from the standard accelerated schemes, because the latter either first update the primal variables $(x, x')$ and then extrapolate the dual variables $(z, z')$ [65] or simultaneously update them both [69, 57], whereas (APGD), which is fashioned along the lines of [81], first updates $(z, z')$ using proximal[3] step and then extrapolates these to update $(x, x')$. Advantage of [81] over the standard rule are three fold; former only needs one proximal step per variable (as opposed to two in [69, 57]) per iteration (which makes it practically faster), or keeps the dual and middle iterates $z_k$, $y_k$ feasible (as opposed to [65, (2.2.17)]), and can easily handle stochastic FO and constraints [53]. Another reason for the choice, which will be evident later on, is that, our update rule can simultaneously provide the optimal complexity for the smooth $\psi_\lambda$ and nonsmooth $f$ parts of the composite function, $\Psi_\lambda$ (5) [55].

With the right choice of $\beta_k$, $\gamma_k$, (APGD) can find an $\varepsilon$-approximate solution to the problem (5), $(x_K, x_K')$, in $\mathcal{O}(\sqrt{2/\lambda\varepsilon})$ steps. Now if we choose $\lambda = \mathcal{O}(\varepsilon)$, we can show that $x_K$ is also an $\mathcal{O}(\varepsilon)$ solution of our original nonsmooth constrained problem (1). This is formalized in the Lemma 2. Thus applying (APGD) on (5) with $\lambda = \varepsilon/G^2$ gives us an $\varepsilon$ solution to the original problem (1) using only $K = \mathcal{O}(G/\varepsilon)$ projections, which is a significant improvement over the $\mathcal{O}(G^2/\varepsilon^2)$ PO calls used by the standard subgradient method.

For a general $G$-Lipschitz convex function $f$, we cannot solve $\text{prox}_{f/\beta_k}$ exactly, and hence we resort to some approximate solution. We emphasize here that it is not immediately evident that we can implement an inexact prox operator, and still maintain that the total number of FO calls used by this inexact APGD method match the optimal lowerbound $\mathcal{O}(G^2/\varepsilon^2)$ [64, 65]. Perhaps surprisingly, this is achieved using the Gradient Sliding method [55], which proposes a specific form of an inexact APGD. Note that the constrained variable $x \in \mathcal{X}$ is not an input to the nonsmooth part $f$ of $\Psi_\lambda$, which means that approximately resolving the prox operator $\text{prox}_{f/t}$ does not require any projection.

As an intermediate algorithm we first present (IAPGD), which is derived from (APGD) but replaces the proximal update of $z_k'$ with an inexact resolution for prox operator $\text{prox}_{f/\beta_k}$ up to an approximation error of $\delta$. Notice that $\delta = 0$, implies that $z_k$ is an exact resolution of the operation $\text{prox}_{f/\beta_k}(\hat{z}_k')$.

The specific choice of the approximation error is important, as other notions of approximation error of the proximal operator in the context of APGD (such as those in [77]) do not explicitly control the distance $\|x' - z'_k\|^2$ which is crucial for our guarantee. Although with $\delta = \varepsilon$, (IAPGD) would require $\mathcal{O}(1/\varepsilon^3)$ FO calls, we provide the details of its analysis, in the next theorem, as it showcases some of the ideas behind the design of our our main algorithm (Algorithm 4).

---

Use the update rule of (APGD), but replace $\mathrm{prox}$ step by the following:

find $z'_k$ satisfying the following for all $x' \in \mathcal{X}'$

$$\frac{\beta_k}{2}\|x' - z'_k\|^2 + f(z'_k) + \frac{\beta_k}{2}\|z'_k - \hat{z}'_k\|^2 \leq f(x') + \frac{\beta_k}{2}\|x' - \hat{z}'_k\|^2 + \delta \ ,$$ (IAPGD)

where $\hat{z}'_k = (z'_{k-1} - \nabla_{x'}\psi_\lambda(y_k, y'_k)/\beta_k)$

---

**Theorem 3.** *Let $f : \mathcal{X}' \to \mathbb{R}$ be a $G$-Lipschitz continuous convex function and $\mathcal{X} \subseteq \mathcal{X}'$ be any convex set with a diameter $D_\mathcal{X}$ and projection oracle $\mathcal{P}_\mathcal{X}$. If we choose $\lambda = \varepsilon/G^2$ and $\delta = \varepsilon$, then after $K = \mathcal{O}(GD_\mathcal{X}/\varepsilon)$ iterations of the IAPGD update rule, initialized with $y'_0 = y_0 = x'_0 = x_0$, finds $x_K \in \mathcal{X}$ satisfying $f(x_K) - \min_{x \in \mathcal{X}} f(x) \leq \varepsilon$. Further, $\mathcal{O}(G^2 D_\mathcal{X}^2/\varepsilon^2)$ iterations of a standard subgradient method ensures the condition in (IAPGD). In total, this algorithm requires $\mathcal{O}(G^3 D_\mathcal{X}^3/\varepsilon^3)$ FO calls and $\mathcal{O}(GD_\mathcal{X}/\varepsilon)$ PO calls.*

**Remarks:** Even though IAPGD only achieves a FO-CC of $\mathcal{O}(1/\varepsilon^3)$, the main take away from this result should be that with this right choice for approximate resolvent of the proxoperator $\mathrm{prox}_{f/\beta_k}$ (IAPGD), we can achieve $\mathcal{O}(1/\varepsilon)$ PO-CC. This is exploited by our MOPES method (Algorithm 1) which uses a more efficient Prox-Slide procedure [55] to approximately resolve the prox operator, so as to obtain a PO-CC of $\mathcal{O}(1/\varepsilon)$ while still maintaining the optimal FO-CC o $\mathcal{O}(1/\varepsilon^2)$.

*Proof of Theorem 3.* Now consider the following potential (Lyapunov) function from [6] for arbitrary $x \in \mathcal{X}$ and $x' \in \mathcal{X}'$:

$$\Phi_k := k(k+1)(\Psi_\lambda(x_k, x'_k) - \Psi_\lambda(x, x')) + (4/\lambda)\|(z_k, z'_k) - (x, x')\|^2 \tag{11}$$

We will prove that this potential satisfy the following approximate descent condition $\Phi_k \leq \Phi_{k-1} + k\varepsilon$ as follows. Notice that by $2/\lambda$-smoothness and convexity of $\psi_\lambda$

$$\psi_\lambda(x_k, x'_k) \leq \psi_\lambda(y_k, y'_k) + \langle \nabla_k, (x_k, x'_k) - (y_k, y'_k) \rangle + \frac{1}{\lambda}\|(x_k, x'_k) - (y_k, y'_k)\|^2$$

$$\leq (1 - \gamma_k)\psi_\lambda(x_{k-1}, x'_{k-1}) +$$

$$\gamma_k[\psi_\lambda(y_k, y'_k) + \langle \nabla_k, (z_k, z'_k) - (y_k, y'_k) \rangle + \frac{\gamma_k}{\lambda}\|(z_k, z'_k) - (z_{k-1}, z'_{k-1})\|^2] \tag{12}$$

where we use the shorthand $\nabla_k := [\nabla_{k,x}^T \nabla_{k,x'}^T]^T := [\nabla_x \psi_\lambda(y_k, y'_k)^T \ \nabla_{x'}\psi_\lambda(y_k, y'_k)^T]^T$. Now combining this with $f(x'_k) \leq (1 - \gamma_k)f(x'_{k-1}) + \gamma_k f(z'_k)$ (convexity) and $\gamma_k/\lambda \leq \beta_k/2$ we get that

$$k(k+1)\Psi_\lambda(x_k, x'_k)$$

$$\leq k(k-1)\Psi_\lambda(x_{k-1}, x'_{k-1}) + 2k\psi_\lambda(y_k, y'_k) + 2k[\langle \nabla_{k,x}, z_k - y_k \rangle + \frac{\beta_k}{2}\|z_k - z_{k-1}\|^2]$$

$$2k[f(z'_k) + \langle \nabla_{k,x'}, z'_k - y'_k \rangle + \frac{\beta_k}{2}\|z'_k - z'_{k-1}\|^2]$$

$$\leq k(k-1)\Psi_\lambda(x_{k-1}, x'_{k-1}) + 2k\psi_\lambda(y_k, y'_k) +$$

$$2k[\langle \nabla_{k,x}, x - y_k \rangle + \frac{\beta_k}{2}(\|z_{k-1} - x\|^2 - \|z_k - x\|^2)]$$

$$2k[f(x') + \langle \nabla_{k,x'}, x' - y'_k \rangle + \frac{\beta_k}{2}(\|z'_{k-1} - x'\|^2 - \|z'_k - x'\|^2) + \varepsilon]$$

$$\leq k(k-1)\Psi_\lambda(x_{k-1}, x'_{k-1}) + 2k\Psi_\lambda(x, x') +$$

$$(4/\lambda)(\|(z_{k-1}, z'_{k-1}) - (x, x')\|^2 - \|(z_k, z'_k) - (x, x')\|^2) + k\varepsilon \ , \tag{13}$$

where the second inequality uses the definition of projection and the $\varepsilon$-approximate resolution of the proximal operator (IAPGD), and the last inequality again uses convexity of $\psi_\lambda$. This proves that

$\Phi_k \leq \Phi_{k-1} + k\varepsilon$, which directly implies that

$$\Psi_\lambda(x_K, x'_K) - \Psi_\lambda(x, x') \leq \frac{4(\|x_0 - x\|^2 + \|x_0 - x'\|^2)}{\lambda K(K+1)} + \frac{1}{K(K+1)}\sum_{k=1}^{K} k\varepsilon \qquad (14)$$

Setting $x' = x$, choosing $\lambda = \varepsilon/G^2$ and $K = \mathcal{O}(GD_\mathcal{X}/\varepsilon)$ gives us $\Psi_\lambda(x_k, x'_k) - f(x) \leq \varepsilon/2$. Then by Lemma 2 we get that $f(x_k) - \min_{x \in \mathcal{X}} f(x) \leq \varepsilon$. For each inner problem the standard (unconstrained) proximal subgradient method applied on $\min_{x' \in \mathcal{X}'} f(x') + (\beta_k/2)\|x' - (z'_{k-1} - \nabla_{k,x'}/\beta_k)\|^2$, initialized with $x_0$ (for ease of argument), can achieve this error using $\mathcal{O}(G^2\|x_0 - \widehat{x}_\lambda(x)\|^2/\varepsilon^2) = \mathcal{O}(G^2 D_\mathcal{X}^2/\varepsilon^2)$ FO calls (Lemma 3, in Appendix B.1). Thus the algorithm uses totally $\mathcal{O}(GD_\mathcal{X}/\varepsilon)$ projections and $\mathcal{O}(G^3 D_\mathcal{X}^3/\varepsilon^3)$ subgradients. □

## A.2 A proof sketch for Theorem 1 (MOPES)

This section provides a short proof sketch for the Theorem 1—guarantee for the MOPES (Algorithm 1) method—to showcase the main analysis techniques used by the full proof in Appendix C.2.1. At a high level, our proof uses a potential function [6] for analyzing APGD, combines it with Proposition 3 which provides a fast convergence guarantee on `Prox-Slide` iterates, and then apply standard APGD proof techniques [81] to obtain the final result.

*Proof sketch.* Here we only consider the deterministic FO. We define the following potential (Lyapunov) function for some arbitrary $x \in \mathcal{X}$, by slightly modifying the standard AGD potential [6].

$$\Phi_k := k(k+1)(\Psi_\lambda(x_k, x'_k) - \Psi_\lambda(x, x)) + (4/\lambda)(\|z_k - x\|^2 + \tau_{k+1}\|z'_k - x\|^2) \qquad (15)$$

where $\tau_k := (T_k + 1)(T_k + 2)/T_k(T_k + 3)$. We will prove that this potential satisfies the descent rule: $\Phi_k \leq \Phi_{k-1} + k\eta'_k$, for some error $\eta'_k$. Using the fact that $\Psi_\lambda$ is a sum of two convex functions: $2/\lambda$-smooth quadratic $\psi_\lambda$ and $G$-Lipschitz $f$, and standard analysis techniques for AGD we can get

$$k(k+1)\Psi_\lambda(x_k, x'_k) \leq k(k-1)\Psi_\lambda(x_{k-1}, x'_{k-1}) + 2k\psi_\lambda(x, x)+$$
$$2k[\langle \nabla_{k,x}, z_k \rangle + (\beta_k/2)\|z_k - z_{k-1}\|^2] + 2k[\phi_k(\widetilde{z}'_k) - \phi_k(x) + (\beta_k/2)\|x - z'_{k-1}\|^2] \qquad (16)$$

where we use the short-hands $\nabla_k := \nabla\psi_\lambda(y_k, y'_k)$ and $\phi_k(x') := f(x') + \langle \nabla_{k,x'}, x' \rangle + (\beta_k/2)\|x' - z'_{k-1}\|^2$. Next, using definition of projection $z_k$, we bound the third term in the RHS of (16) as

$$2k[\langle \nabla_{k,x}, x - z_k \rangle + (\beta_k/2)\|z_k - z_{k-1}\|^2] \leq 2k(\beta_k/2)[\|z_{k-1} - x\|^2 - \|z_k - x\|^2]. \qquad (17)$$

The fourth term in the RHS of (16) corresponds to the $\varepsilon$-approximate resolution of the $\text{prox}_{f/\beta_k}$ operator through the `Prox-Slide` procedure (Line 1.5), whose output satisfies the following guarantee.

**Proposition 2** (informal version of Proposition 3). *Output of `Prox-Slide` satisfies*

$$\phi_k(\widetilde{z}'_k) - \phi_k(x) + \frac{\beta_k}{2}\|z'_k - x\|^2 \leq \frac{\beta_k}{2}(\tau_k - 1)[\|z'_{k-1} - x\|^2 - \|z'_k - x\|^2] + \frac{16\,G^2}{\beta_k T_k}.$$

The above lemma guarantees the optimal $O(1/T_k)$ convergence rate for the strongly convex minimization problem: $\min_{z' \in \mathcal{X}'} \phi_k(z')$, corresponding to the proximal operator. By combining the inequalities (16) and (17) and the proposition we get: $\Phi_k \leq \Phi_{k-1} + k\mathcal{O}(G^2/\beta_k T_k)$. Now, using Lemma 2, and setting $x = x^*$, $\lambda = \frac{\varepsilon}{G^2}$, $\beta_k = \frac{4}{\lambda k}$, $T_k = \mathcal{O}(k)$ and $K = \Theta(\frac{G\|x_0 - x^*\|}{\varepsilon})$ we get

$$f(x_K) - f(x^*) \leq \Psi_\lambda(x_K, x'_K) - \Psi_\lambda(x^*, x^*) + G^2\frac{\lambda}{2}$$

$$\leq \frac{8\|x_0 - x^*\|^2}{\lambda K(K+1)} + \frac{\sum_{k=1}^{K} k\, 16G^2/\beta_k T_k}{\lambda K(K+1)} + G^2\frac{\lambda}{2} = \mathcal{O}(\varepsilon)$$

Therefore, the total number of PO calls made is $K = \mathcal{O}(G\|x_0 - x^*\|/\varepsilon)$ and the total number of FO calls made is $\sum_{k=1}^{K} T_k = \mathcal{O}(K^2) = \mathcal{O}(G^2\|x_0 - x^*\|^2/\varepsilon^2)$. □

# B   Supporting results

## B.1   Proximal Subgradient method

**Lemma 3** (proximal subgradient descent). *Consider the regularized optimization problem*

$$\min_u [f_{\beta,x}(u) := f(u) + (\beta/2)\|u - x\|^2] \qquad (18)$$

*and the proximal subgradient method's update rule*

$$u_{t+1} = \operatorname{argmin}_u [F_t(u) := \langle g_t, u - x \rangle + (1/2\eta)\|u - u_t\|^2 + \beta/2\|u - x\|^2]$$
$$= u_t - (\eta/(1 + \eta\beta))(g_t + \beta(u_t - x)) \qquad (19)$$

*where $g_t \in \partial f(u_t)$ and $\eta$ is the effective stepsize. Now, if $\eta = 2\,G^2\|u_0 - u\|/\sqrt{T}$ and $\widetilde{u}_T = \frac{1}{T}\sum_{t=0}^{T-1} u_{t+1}$, then for any $u$*

$$\frac{\beta}{2}\|\widetilde{u}_T - u\|^2 + f_{\beta,x}(\widetilde{u}_T) - f_{\beta,x}(u) \leq \frac{2\,G\,\|u_0 - u\|}{\sqrt{T}} \qquad (20)$$

*Proof.* Let $u$ be an arbitrary feasible point. By convexity and $G$-Lipschitzness of $f$,

$$f(u_{t+1}) - f(u) = f(u_{t+1}) - f(u_t) + f(u_t) - f(u)$$
$$\leq \langle g_{t+1}, u_{t+1} - u_t \rangle + \langle g_t, u_t - u \rangle$$
$$= \langle g_t, u_{t+1} - u_t \rangle + \langle g_{t+1} - g_t, u_{t+1} - u_t \rangle + \langle g_t, u_t - u \rangle$$
$$\leq \langle g_t, u_{t+1} - u \rangle + 2G\,\|u_{t+1} - u_t\|, \qquad (21)$$

As $u_{t+1}$ is the minimizer of a $(\beta + 1/\eta)$-strong convexity update objective $F_t$ and since, we get that

$$\left(\frac{\beta}{2} + \frac{1}{2\eta}\right)\|u_{t+1} - u\|^2 + F_t(u_{t+1}) \leq F_t(u) \qquad (22)$$

Now summing up (21), sand (22) we get

$$\frac{\beta}{2}\|u_{t+1} - u\|^2 + f_{\beta,x}(u_{t+1}) - f_{\beta,x}(u) \leq \frac{1}{2\eta}(\|u_t - u\|^2 - \|u_{t+1} - u\|^2) +$$
$$2G\,\|u_{t+1} - u_t\| - \frac{1}{2\eta}\|u_{t+1} - u_t\|^2$$
$$\leq \frac{1}{2\eta}(\|u_t - u\|^2 - \|u_{t+1} - u\|^2) + 2\,G^2\eta$$
$$\implies \frac{1}{T}\sum_{t=0}^{T-1}\frac{\beta}{2}\|u_{t+1} - u\|^2 + f_{\beta,x}(u_{t+1}) - f_{\beta,x}(u) \leq \frac{1}{2\eta T}(\|u_0 - u\|^2 - \|u_T - u\|^2) + 2\,G^2\eta$$
$$\frac{\beta}{2}\|\widetilde{u}_T - u\|^2 + f_{\beta,x}(\widetilde{u}_T) - f_{\beta,x}(u) \leq \qquad (23)$$

where the second inequality follows from $ax - x^2/2b \leq a^2 b/2$, the third inequality is obtained by summing over $t = 0, \ldots, T - 1$, and the third inequality uses Jensen's inequality. Choosing $T = 2\,G\,\|u_0 - u\|/\sqrt{T}$, we get the desired result

$$\frac{\beta}{2}\|\widetilde{u}_T - u\|^2 + f_{\beta,x}(\widetilde{u}_T) - f_{\beta,x}(u) \leq \frac{2\,G\,\|u_0 - u\|}{\sqrt{T}} \qquad (24)$$

□

## B.2   Frank-Wolfe projected subgradient method FW-PGD (Algorithm 3)

Here we provide the details of the Frank-Wolfe based projected subgradient method (Algorithm 3) used in the experiments. The main idea is to use some competitive LMO based method to approximate the projection step in the standard projected subgradient method. The following theorem gives some guarantees for the output of the Algorithm 3.

**Algorithm 3:** Frank-Wolfe projected subgradient method using LMO

---

**Input:** $f$, $\mathcal{X}$, $G$, $D_{\mathcal{X}}$, $x_0$, $K$,

**3.1** **for** $k = 0, \ldots, K - 1$ **do**

**3.2** $\quad$ Set $\widehat{g}_k = \mathrm{SFO}\,(x_k)$

**3.3** $\quad$ Using any competitive LMO based algorithm (e.g. Frank-Wolfe method [28] or $\mathrm{CndG}$ procedure [56, Algo. 1]), approximately solve the projection problem

$$x_{k+1} \approx \operatorname*{argmin}_{x \in \mathcal{X}} \langle \widehat{g}_k, x \rangle + \frac{1}{2\alpha_k} \|x - x_k\|^2 = \operatorname*{argmin}_{x \in \mathcal{X}} \frac{1}{2\alpha_k} \|x - (x_k - \alpha_k \cdot \widehat{g}_k)\|^2 , \quad (25)$$

$\quad$ ensuring that the Wolfe duality gap [45] of the above problem at $u_{\Pi}$ satisfies

$$x \max_{s \in \mathcal{X}} \langle \widehat{g}_k + 1/\alpha_k\,(x_{k+1} - x_k)\,, x_{k+1} - s \rangle \leq \eta_k \quad (26)$$

**Output:** $\bar{x}_K = \frac{\sum_{k=0}^{K-1} \alpha_k x_k}{\sum_{k=0}^{K-1} \alpha_k}$

---

**Theorem 4.** *Let $f : \mathcal{X}' \to \mathbb{R}$ be a $G$-Lipschitz continuous proper l.s.c. convex function, and $\mathcal{X} \subseteq \mathcal{X}'$ be some closed convex subset of $\mathbb{R}^d$ with diameter $D_{\mathcal{X}}$. Then after $K$ iterations, the Algorithm 3 projection tolerance $\eta_k = (G^2 + \sigma^2)\alpha_k$, stepsize $\alpha_k = \frac{D_{\mathcal{X}}}{2\sqrt{G^2+\sigma^2}\sqrt{K}}$ and outputs $\bar{x}_K \in \mathcal{X}$ satisfying*

$$\mathbb{E}[f\,(\bar{x}_K)] - f(x^*) \leq \frac{2\sqrt{G^2 + \sigma^2}D_{\mathcal{X}}}{\sqrt{K}} \quad (27)$$

*Further, the algorithm uses $K$ SFO calls and $O(K^2)$ LMO calls.*

*Proof.* Using the Wolfe duality gap guarantee we get that for any $x \in \mathcal{X}$

$$\left\langle \widehat{g}_k + \frac{1}{\alpha_k}\,(x_{k+1} - x_k)\,, x_{k+1} - x \right\rangle \leq \eta_k . \quad (28)$$

By rearranging the terms above we get that

$$\langle \widehat{g}_k, x_k - x \rangle \leq \frac{1}{2\alpha_k}(\|x_k - x\|^2 - \|x_{k+1} - x\|^2) - \frac{1}{2\alpha_k}\|x_{k+1} - x_k\|^2 + \langle \widehat{g}_k, x_k - x_{k+1} \rangle + \eta_k$$

$$\leq \frac{1}{2\alpha_k}(\|x_k - x\|^2 - \|x_{k+1} - x\|^2) - \frac{1}{2\alpha_k}\|x_{k+1} - x_k\|^2 + \|\widehat{g}_k\|\|x_k - x_{k+1}\| + \eta_k$$

$$\leq \frac{1}{2\alpha_k}(\|x_k - x\|^2 - \|x_{k+1} - x\|^2) + \frac{\alpha_k}{2}\|\widehat{g}_k\|^2 + \eta_k , \quad (29)$$

where the last inequality uses the fact that $-(a/2)z^2 + bz \leq b^2/2a$ for all $a, b, z \in \mathbb{R}$. Next, multiplying by $\alpha_k$ and summing the above inequality over $k = 0, \ldots, K - 1$ and dividing by $\sum_{k'=0}^{K-1} \alpha_{k'}$ we get

$$\sum_{k=0}^{K-1} \alpha_k \langle \widehat{g}_k, x_k - x \rangle \leq \frac{1}{2}(\|x_0 - x\|^2 - \|x_K - x\|^2) + \sum_{k=0}^{K-1} \alpha_k^2 (\frac{\|\widehat{g}_k\|^2}{2} + \frac{\eta_k}{\alpha_k}), \quad (30)$$

Now taking expectation w.r.t. all the stochasticity in $\{\widehat{g}_k\}_{k=0}^{K-1}$ on both sides and using, the towering conditional expectation property $\mathbb{E}[a] = \mathbb{E}[\mathbb{E}[a\,|\,x_k]]$, and $\mathbb{E}[\widehat{g}_k\,|\,x_k] = g_k \in \partial f(x_k)$ and $\mathbb{E}[\|\widehat{g}_k\|^2\,|\,x_k] \leq 2(G^2 + \sigma^2)$ we get

$$\sum_{k=0}^{K-1} \alpha_k \mathbb{E}[\langle g_k, x_k - x \rangle] \leq \frac{1}{2}\|x_0 - x\|^2 + \sum_{k=0}^{K-1} \alpha_k^2((G^2 + \sigma^2) + \frac{\eta_k}{\alpha_k}), \quad (31)$$

Next diving by $\sum_{k'=0}^{K-1}\alpha_{k'}$, using convex affine lower bound of $f$ at $x_k$ and Jensen's inequality we get

$$\sum_{k=0}^{K-1}\frac{\alpha_k}{\sum_{k'=0}^{K-1}\alpha_{k'}}\mathbb{E}[f(x_k)-f(x)]\leq\frac{\frac{1}{2}\|x_0-x\|^2+\sum_{k=0}^{K-1}\alpha_k^2((G^2+\sigma^2)+\frac{\eta_k}{\alpha_k})}{\sum_{k'=0}^{K-1}\alpha_{k'}}$$

$$\mathbb{E}\left[f\left(\sum_{k=0}^{K-1}\frac{\alpha_k\cdot x_k}{\sum_{k'=0}^{K-1}\alpha_{k'}}\right)\right]-f(x)\leq \tag{32}$$

Next if we choose $\eta_k=\alpha_k(G^2+\sigma^2)$, and set $x=x^*\in\operatorname{argmin}_{x'\in\mathcal{X}}f(x')$ and $\alpha_k=\frac{D_{\mathcal{X}}}{2\sqrt{G^2+\sigma^2}\sqrt{K}}$ we get

$$\mathbb{E}\left[f\left(\sum_{k=0}^{K-1}\frac{\alpha_k\cdot x_k}{\sum_{k'=0}^{K-1}\alpha_{k'}}\right)\right]-f(x^*)\leq\frac{\frac{1}{2}D_{\mathcal{X}}^2+\sum_{k=0}^{K-1}\alpha_k^2\cdot 2(G^2+\sigma^2)}{\sum_{k'=0}^{K-1}\alpha_{k'}}$$

$$=\frac{2\sqrt{G^2+\sigma^2}D_{\mathcal{X}}}{\sqrt{K}} \tag{33}$$

Clearly the algorithm uses $K$ SFO calls. At step $k$ when approximating the projection using an LMO based method, after using $\hat{T}_k=\lceil\frac{7D_{\mathcal{X}}^2}{\alpha_k^2(G^2+\sigma^2)}\rceil$ LMO calls in the `Approx-Proj` procedure, the Wolfe duality gap (38) is at most $\lceil\frac{6(1/\alpha_k)D_{\mathcal{X}}^2}{\hat{T}_k}\rceil\leq\alpha_k(G^2+\sigma^2)$ if we use CndG procedure [56, Theorem 2.2(c)] or $\lceil\frac{7(1/\alpha_k)D_{\mathcal{X}}^2}{\hat{T}_k}\rceil\leq\alpha_k(G^2+\sigma^2)$ if we use the standard Frank-Wolfe algorithm [45, Theorem 2]. Therefore the total number of linear minimization oracle calls made by the algorithm is

$$\sum_{k=0}^{K-1}\hat{T}_k=\sum_{k=0}^{K-1}\frac{7D_{\mathcal{X}}^2}{\alpha_k^2(G^2+\sigma^2)}+K=28K^2+K=O(K^2) \tag{34}$$

where we use the given choice for $\alpha_k=\frac{D_{\mathcal{X}}}{2\sqrt{G^2+\sigma^2}\sqrt{K}}$. $\qquad\square$

## C  Proofs of the main results

### C.1  Proof of Lemma 2

*Proof.* First we prove part $(i)$. By definitions of $\Psi_\lambda$ (5) and $f_\lambda$ (Definition 3), we have $\min_{x\in\mathcal{X}}\min_{x\in\mathcal{X}'}\Psi_\lambda(x,x')=\min_{x\in\mathcal{X}}f_\lambda(x)$. By Lemma 1(a), we also can show that $\min_{x\in\mathcal{X}}f_\lambda(x)\leq\min_{x\in\mathcal{X}}f(x)$.

For part $(ii)$, first we show the following.

$$\mathbb{E}f_\lambda(x_\varepsilon)=\mathbb{E}\Psi_\lambda(x_\varepsilon,\widehat{x}_\lambda(x_\varepsilon))=\mathbb{E}\min_{x'(x_\varepsilon)}\Psi_\lambda(x_\varepsilon,x'(x_\varepsilon))\leq\mathbb{E}_{\bar{x}_K}\Psi_\lambda(x_\varepsilon,\mathbb{E}_{x'_\varepsilon|x_\varepsilon}x'_\varepsilon)$$

$$\leq\mathbb{E}_{x_\varepsilon}\mathbb{E}_{x'_\varepsilon|x_\varepsilon}\Psi_\lambda(x_\varepsilon,x'_\varepsilon)$$

$$=\mathbb{E}\Psi_\lambda(x_\varepsilon,x'_\varepsilon) \tag{35}$$

Finally, combining the above inequality with Lemma 1(c) we get the desired result

$$\mathbb{E}f(x_\varepsilon)\leq\mathbb{E}f_\lambda(x_\varepsilon)+G^2\lambda/2\leq\mathbb{E}\Psi_\lambda(x_\varepsilon,x'_\varepsilon)+G^2\lambda/2 \tag{36}$$

$\qquad\square$

### C.2  Analysis of MOPES (Algorithm 1) and MOLES (Algorithm 2) method

Instead of separately analyzing MOPES and MOLES, we first analyze a more general algorithm, Algorithm 4, which has the following guarantee.

**Theorem 5.** *Let $f:\mathcal{X}'\to\mathbb{R}$ be a $G$-Lipschitz continuous proper l.s.c. convex function, and $\mathcal{X}\subseteq\mathcal{X}'=B(0,R)$ be some convex subset contained inside the Euclidean ball of radius $R$ around origin. Then after $K$ iterations, the Algorithm 4 outputs $x_K\in\mathcal{X}$ satisfying*

$$\mathbb{E}[f(x_K)]-f(x^*)\leq\frac{10\|x_0-x^*\|^2+8\tilde{D}}{\lambda K(K+1)}+\frac{\sum_{k=1}^{K}2k\,\eta_k}{K(K+1)}+G^2\frac{\lambda}{2} \tag{39}$$

*for any choice of $\lambda>0$, $\tilde{D}>0$, and tolerance $\{\eta_k\}_{k\in[K]}$ (38).*

**Algorithm 4:** Moreau subgradient method for nonsmooth convex optimization using PO or LMO

---

**Input:** $f, \mathcal{X}, \mathcal{X}', G, D_{\mathcal{X}}, R, x_0, K, \tilde{D}, \lambda, \{\eta_k \in \mathbb{R}_+\}_{k \in [K]}$

**4.1** Set $x'_0 = z'_0 = x_0 = z_0 = x_0$

**4.2** **for** $k = 1, \ldots, K$ **do**

**4.3** $\quad$ Set $\lambda_k = \lambda$, $\beta_k = \frac{4}{\lambda_k k}$ , $\gamma_k = \frac{2}{k+1}$ , and $T_k = \left\lceil \frac{(4G^2 + \sigma^2)\lambda^2 K k^2}{2\tilde{D}} \right\rceil$

**4.4** $\quad$ Set $(y_k, y'_k) = (1 - \gamma_k) \cdot (x_{k-1}, x'_{k-1}) + \gamma_k \cdot (z_{k-1}, z'_{k-1})$

**4.5** $\quad$ Set $z_k = $ `Approx-Proj` $\left(\nabla_{y_k} \Psi_\lambda(y_k, y'_k), z_{k-1}, \beta_k, \eta_k\right)$ // Note $\nabla_{y_k} \Psi_\lambda(y_k, y'_k) = \frac{y_k - y'_k}{\lambda}$

**4.6** $\quad$ Set $(z'_k, \widetilde{z}'_k) = $ `Prox-Slide`$\left(\nabla_{y'_k} \psi_\lambda(y_k, y'_k), z'_{k-1}, \beta_k, T_k\right)$ $\quad$ // $\nabla_{y'_k} \psi_\lambda(y_k, y'_k) = \frac{y'_k - y_k}{\lambda}$

**4.7** $\quad$ Set $(x_k, x'_k) = (1 - \gamma_k) \cdot (x_{k-1}, x'_{k-1}) + \gamma_k \cdot (z_k, \widetilde{z}'_k)$

$\quad$ **Output:** $(x_K, x'_K)$

**4.8** `Approx-Proj`($g$, $u_0$, $\beta$, $\eta$) *// Approx. resolve $\mathcal{P}_{\mathcal{X}}(u_0 - g/\beta)$[56]*:

**4.9** $\quad$ Either using exact PO, $\mathcal{P}_{\mathcal{X}}$ (1) , or using any competitive LMO based algorithm (e.g. Frank-Wolfe method [28] or $\mathrm{CndG}$ procedure [56, Algo. 1]), approximately solve the projection problem

$$u_\Pi \approx \underset{u \in \mathcal{X}}{\operatorname{argmin}} \langle g, u \rangle + \frac{\beta}{2}\|u - u_0\|^2 = \underset{u \in \mathcal{X}}{\operatorname{argmin}} \frac{\beta}{2}\|u - (u_0 - g/\beta)\|^2, \qquad (37)$$

$\quad$ ensuring that the Wolfe duality gap [45] of the above problem at $u_\Pi$ satisfies

$$\max_{s \in \mathcal{X}} \langle g + \beta (u_\Pi - u_0), u_\Pi - s \rangle \leq \eta_k \qquad (38)$$

$\quad$ **return** $u_\Pi$

**4.10** `Prox-Slide`($g$, $u_0$, $\beta$, $T$) *// Approx. resolve $\mathrm{prox}_{f/\beta}(u_0 - g/\beta)$[55]*:

**4.11** $\quad$ Set $\widetilde{u}_0 = u_0$

**4.12** $\quad$ **for** $t = 1, \ldots, T$ **do**

**4.13** $\quad\quad$ Set $\theta_t = \frac{2(t+1)}{t(t+3)}$, $\widehat{g}_{t-1} = \mathrm{SFO}(u_{t-1})$ (3)

**4.14** $\quad\quad$ Set $\widehat{u}_t = u_{t-1} - \frac{1}{(1+t/2)\beta} \cdot (\widehat{g}_{t-1} + \beta(u_{t-1} - (u_0 - g/\beta)))$

$\quad\quad$ // subgradient method step for $\phi(u) := f(u) + \frac{\beta}{2}\|u - (u_0 - \frac{g}{\beta})\|^2$

**4.15** $\quad\quad$ Set $u_t = \widehat{u}_t \cdot \min(1, R/\|u_t\|)$ $\quad$ // projection of $\widehat{u}_t$ onto $\mathcal{X}'$: $\mathcal{P}'_{\mathcal{X}}(\mathbf{u}_t)$

**4.16** $\quad\quad$ Set $\widetilde{u}_t = (1 - \theta_t) \cdot \widetilde{u}_{t-1} + \theta_t \cdot u_t$

**4.17** $\quad$ **return** $(u_T, \widetilde{u}_T)$

---

**Remarks**: Before providing a proof for the above result we discuss some its implications. MOPES makes $K$ PO calls, one per outer step, and $\sum_{k=1}^{K} T_k = \mathcal{O}(\lambda^2 K^4)$ SFO calls, one per inner step. The above analysis shows that we need to choose $\lambda = \varepsilon/G^2$, which is expected from Lemma 1. Since PO returns exact projections, the second term is zero with $\eta_k = 0$. The target accuracy of $\varepsilon$ is achieved by tuning the first term, where we need to choose $K = \Theta(1/\sqrt{\lambda\varepsilon})$. Put together, this gives the desired $\mathcal{O}(\varepsilon^{-1})$ PO-CC and $\mathcal{O}(\varepsilon^{-2})$ SFO-CC for MOPES. A complete proof is provided in Section C.2.1.

When we have inexact projections in MOLES, we need $\eta_k = \Theta(1/k)$ to ensure that the second term is $\mathcal{O}(\varepsilon)$. At (outer) iteration $k$, this uses $\hat{T} = \Omega(K)$ iterations of Frank-Wolfe algorithm in `FW-Based-Projection` of Algorithm 2. MOLES makes $\sum_{k=1}^{K} \mathcal{O}(K) = \mathcal{O}(K^2)$ LMO calls, resulting in $\mathcal{O}(\varepsilon^{-2})$ LMO-CC as $K = \mathcal{O}(1/\sqrt{\lambda\varepsilon}) = \mathcal{O}(1/\varepsilon)$. A complete proof is in Section C.2.2.

*Proof of Theorem 5.* We define the following potential (Lyapunov) function, for some arbitrary $x \in \mathcal{X}$, $x' \in \mathcal{X}'$:

$$\Phi_k := k(k+1)(\Psi_\lambda(x_k, x'_k) - \Psi_\lambda(x, x')) + \frac{4}{\lambda}(\|z_k - x\|^2 + \frac{(T_{k+1} + 1)(T_{k+1} + 2)}{T_{k+1}(T_{k+1} + 3)}\|z'_k - x'\|^2) \qquad (40)$$

This is a slightly modified version of the following potential function for the standard AGD setting with a $2/\lambda$-smooth function $\Psi_\lambda$ [6]: $k(k+1)(\Psi_\lambda(x_k, x'_k) - \Psi_\lambda(x, x')) + \frac{4}{\lambda}(\|z_k - x\|^2 + \|z'_k - x'\|^2)$. Notice that the modification factor

$$\frac{(T_{k+1} + 1)(T_{k+1} + 2)}{T_{k+1}(T_{k+1} + 3)} \leq \frac{3}{2} = \mathcal{O}(1) \tag{41}$$

is upper-bounded by a constant when $1 \leq T_k$. Below we prove that this potential satisfies the approximate descent guarantee: $\Phi_k \leq \Phi_{k-1} + k\eta_k + k\eta'_k$, for some error $\eta'_k$. First, notice that by $2/\lambda$-smoothness and convexity of $\psi_\lambda$

$$\psi_\lambda(x_k, x'_k) \leq \psi_\lambda(y_k, y'_k) + \langle \nabla_k, (x_k, x'_k) - (y_k, y'_k) \rangle + \frac{1}{\lambda}\|(x_k, x'_k) - (y_k, y'_k)\|^2$$

$$= (1 - \gamma_k)[\psi_\lambda(y_k, y'_k) + \langle \nabla_k, (x_{k-1}, x'_{k-1}) - (y_k, y'_k) \rangle]$$

$$\gamma_k[\psi_\lambda(y_k, y'_k) + \langle \nabla_k, (z_k, \tilde{z}'_k) - (y_k, y'_k) \rangle + \frac{\gamma_k}{\lambda}\|(z_k, \tilde{z}'_k) - (z_{k-1}, z'_{k-1})\|^2]$$

$$\leq (1 - \gamma_k)\psi_\lambda(x_{k-1}, x'_{k-1}) +$$

$$\gamma_k[\psi_\lambda(y_k, y'_k) + \langle \nabla_k, (z_k, \tilde{z}'_k) - (y_k, y'_k) \rangle + \frac{\gamma_k}{\lambda}\|(z_k, \tilde{z}'_k) - (z_{k-1}, z'_{k-1})\|^2] \tag{42}$$

where we use the shorthand $\nabla_k := [\nabla_{k,x}^T \nabla_{k,x'}^T]^T := [\nabla_x \psi_\lambda(y_k, y'_k)^T \ \nabla_{x'} \psi_\lambda(y_k, y'_k)^T]^T$, and the second inequality uses Lines 4.4 and 4.7. Now combining this with $f(x'_k) \leq (1 - \gamma_k)f(x'_{k-1}) + \gamma_k f(\tilde{z}'_k)$ (using convexity of $f$ and Line 4.7) and $\gamma_k/\lambda = 2/(\lambda(k+1)) \leq 2/(\lambda k) = \beta_k/2$ (using Line 4.3), and multiplying it with $k(k+1)$ we get that

$$k(k+1)\Psi_\lambda(x_k, x'_k)$$

$$\leq k(k-1)\Psi_\lambda(x_{k-1}, x'_{k-1}) + 2k\psi_\lambda(y_k, y'_k) + 2k[\langle \nabla_{k,x}, z_k - y_k \rangle + \frac{\beta_k}{2}\|z_k - z_{k-1}\|^2]$$

$$2k[f(\tilde{z}'_k) + \langle \nabla_{k,x'}, \tilde{z}'_k - y'_k \rangle + \frac{\beta_k}{2}\|\tilde{z}'_k - z'_{k-1}\|^2]$$

$$= k(k-1)\Psi_\lambda(x_{k-1}, x'_{k-1}) + 2k\psi_\lambda(y_k, y'_k) + 2k[\langle \nabla_{k,x}, z_k - y_k \rangle + \frac{\beta_k}{2}\|z_k - z_{k-1}\|^2]$$

$$2k[\phi_k(\tilde{z}'_k) - \phi_k(x') + \frac{\beta_k}{2}\|x' - z'_{k-1}\|^2 + f(x') + \langle \nabla_{k,x'}, x' - y'_k \rangle] \tag{43}$$

where for brevity we use the notation

$$\phi_k(x') := f(x') + \langle \nabla_{k,x'}, x' \rangle + \frac{\beta_k}{2}\left\|x' - z'_{k-1}\right\|^2 . \tag{44}$$

Now using the approximate optimality of $z_k$ through the bound on the Wolfe dual gap (38) we get,

$$\frac{\beta_k}{2}\|z_k - z_{k-1}\|^2 = \frac{\beta_k}{2}\|z_{k-1} - x\|^2 + \beta_k \langle z_k - z_{k-1}, z_k - x \rangle - \frac{\beta_k}{2}\|z_k - x\|^2$$

$$\leq \frac{\beta_k}{2}\|z_{k-1} - x\|^2 + \langle \nabla_{k,x}, x - z_k \rangle + \eta_k - \frac{\beta_k}{2}\|z_k - x\|^2 . \tag{45}$$

When $z_k$ is the exact projection (as in Line 1.5) then the above inequality is satisfied with above $\eta_k = 0$. Otherwise, with the LMO oracle we will later set $\eta_k = \mathcal{O}(\varepsilon)$. Next we state the following lemma which provides a guarantee for the `Prox-Slide` procedure [55]. Here for rigorousness, we denote the iterates of the `Prox-Slide` procedure (Line 4.10) called at the outer step $k$, with $\{u_{k,t}\}_t$. Similarly at the outer step $k$, we denote the stochastic subgradients used by the `Prox-Slide` procedure and the corresponding subgradient with $\{\hat{g}_{k,t}\}_t$ and $\{g_{k,t}\}_t$, i.e. $g_{k,t} := \mathbb{E}[\hat{g}_{k,t}|u_{k,t}] \in \partial f(u_{k,t})$ for all $k$ and $t$. A proof for this lemma is provided in Section C.2.3.

**Proposition 3** ([55, Similar to Proposition 1]). *Let $\phi_k$ (44) be the minimization objective solved by* `Prox-Slide` *procedure at step $k$ of Algorithm 4. Then $(z'_k, \tilde{z}'_k)$ obtained after $T_k$ iterations of the procedure satisfy the following for any $x' \in \mathcal{X}'$,*

$$\phi_k(\tilde{z}'_k) - \phi_k(x') \leq \frac{2}{T_k(T_k + 3)}\frac{\beta_k}{2}\|z'_{k-1} - x'\|^2 - \frac{(T_k + 1)(T_k + 2)}{T_k(T_k + 3)}\frac{\beta_k}{2}\|z'_k - x'\|^2 +$$

$$\frac{4\sum_{t=0}^{T_k-1}(2G + \|\delta_{k,t}\|)^2}{\beta_k T_k(T_k + 3)} + \sum_{t=0}^{T_k-1}\frac{2(t+2)}{T_k(T_k + 3)}\langle \delta_{k,t}, x' - u_{k,t} \rangle \tag{46}$$

*where $\delta_{k,t} := \hat{g}_{k,t} - g_{k,t}$ and $u_{k,t}$ are private inner variable of the* `Prox-Slide` *procedure.*

*Aside:* Note that `Prox-Slide` procedure essentially applies $T_k$ steps of the proximal standard subgradient method to the $\phi_k$ (44), which is a composite function of a $G$-Lipschitz function $f$ and prox-friendly $\beta_k$-strongly convex quadratic. Finally the procedure outputs the average of its iterate $\widetilde{z}'_k$ and its last iterate $z'_k$. In the end we will set $T_k = \Theta(1/\varepsilon)$ and $K = \Theta(1/\varepsilon)$ so that total number of subgradients used by the algorithm be $\sum_{k=1}^{K} T_k = \mathcal{O}(1/\varepsilon^2)$.

Now substituting (45) and Proposition 3 into (43) we get

$$k(k+1)\Psi_\lambda(x_k, x'_k) \le k(k-1)\Psi_\lambda(x_{k-1}, x'_{k-1}) + 2k\psi_\lambda(y_k, y'_k) +$$

$$2k[\langle \nabla_{k,x}, x - y_k \rangle + \frac{\beta_k}{2}(\|z_{k-1} - x\|^2 - \|z_k - x\|^2) + \eta_k]$$

$$2k[f(x') + \langle \nabla_{k,x'}, x' - y'_k \rangle] +$$

$$2k[\frac{(T_k+1)(T_k+2)}{T_k(T_k+3)}\frac{\beta_k}{2}\|z'_{k-1} - x'\|^2 - \frac{(T_k+1)(T_k+2)}{T_k(T_k+3)}\frac{\beta_k}{2}\|z'_k - u\|^2 + \eta'_k]$$

$$\le k(k-1)\Psi_\lambda(x_{k-1}, x'_{k-1}) + 2k\Psi_\lambda(x, x') + 2k(\eta_k + \eta'_k)$$

$$\frac{4}{\lambda}(\|z_{k-1} - x\|^2 - \|z_k - x\|^2)$$

$$\frac{4}{\lambda}(\frac{(T_k+1)(T_k+2)}{T_k(T_k+3)}\|z'_{k-1} - x'\|^2 - \frac{(T_{k+1}+1)(T_{k+1}+2)}{T_{k+1}(T_{k+1}+3)}\|z'_k - x'\|^2),$$

$$\tag{47}$$

where we use the shorthand

$$\eta'_k := \frac{4\sum_{t=0}^{T_k-1}(2G + \|\delta_{k,t}\|)^2}{\beta_k T_k(T_k+3)} + \sum_{t=0}^{T_k-1}\frac{2(t+2)}{T_k(T_k+3)}\langle \delta_{k,t}, x' - u_{k,t} \rangle, \tag{48}$$

and the last inequality uses convexity of $\psi_\lambda$ and $\Psi_\lambda = f + \psi_\lambda$, definition of $\beta_k$ (Line 4.3), and the fact that

$$T_k \le T_{k+1} \quad \text{(Line 4.3)} \quad , \text{ and } \quad \frac{(T_{k+1}+1)(T_{k+1}+2)}{T_{k+1}(T_{k+1}+3)} \le \frac{(T_k+1)(T_k+2)}{T_k(T_k+3)} \tag{49}$$

This proves the approximate descent guarantee: $\Phi_k \le \Phi_{k-1} + k(\eta_k + \eta'_k)$, which along with the facts: $1 \le T_1$ and $z_0 = z'_0 = x_0$ gives

$$\Psi_\lambda(x_K, x'_K) - \Psi_\lambda(x, x') \le \frac{4(\|x_0 - x\|^2 + (3/2)\|x_0 - x'\|^2)}{\lambda K(K+1)} + \frac{\sum_{k=1}^{K} 2k(\eta_k + \eta'_k)}{K(K+1)} \tag{50}$$

Now we take expectation, with respect to randomness in all the stochastic subgradients $((\widehat{g}_{k,i})_{i=1}^{T_k})_{k=1}^{K}$ used in the algorithm, on both sides of (50). Then the expectation of the error from the `Prox-Slide` procedure can be bounded as follows

$$\sum_{k=1}^{K} 2k\mathbb{E}[\eta'_k] = \sum_{k=1}^{K} 2k\mathbb{E}[\frac{4\sum_{t=0}^{T_k-1}(2G + \|\delta_{k,t}\|)^2}{\beta_k T_k(T_k+3)} + \sum_{t=0}^{T_k-1}\frac{2(t+2)}{T_k(T_k+3)}\langle \delta_{k,t}, u - u_t \rangle]$$

$$\le \sum_{k=1}^{K} 2k\frac{8(4G^2 + \sigma^2)}{(\frac{4}{\lambda k})(\frac{(4G^2+\sigma^2)\lambda^2 K k^2}{2\tilde{D}})} + 0$$

$$= \frac{8\tilde{D}}{\lambda} \tag{51}$$

where we use (48), linearity of expectation, $(a+b)^2 \le 2(a^2 + b^2)$, variance of stochastic gradient $\mathbb{E}[\|\delta_{k,t}\|^2 | u_{k,t}] = \mathbb{E}[\|\widehat{g}_{k,t} - g_{k,t}\|^2 | u_{k,t}] \le \sigma^2$ (3), the value of $T_k$ from Line 4.3, and the fact that expectation of the second term becomes zero, since $\mathbb{E}[\widehat{g}_{k,i-1} | u_{k,i-1}] = g_{k,i-1}$, which in turn implies

$$\mathbb{E}[\langle \delta_{k,t}, x' - u_{k,t} \rangle] = \mathbb{E}[\mathbb{E}[\langle \widehat{g}_{k,t} - g_{k,t}, x' - u_{k,t} \rangle | u_{k,t}]]s = \mathbb{E}[\langle 0, x' - u_{k,i-1} \rangle] = 0 . \tag{52}$$

*Aside:* Note that, in the final guarantee, when we set $\lambda = \varepsilon/G^2$ and $K = \mathcal{O}(1/\varepsilon)$, we are setting $T_k = \Theta(\varepsilon k^2) = \mathcal{O}(1/\varepsilon)$ and $1/\beta_k = \Theta(1/\varepsilon k) = \mathcal{O}(1)$, so that the error from the `Prox-Slide` procedure is small enough. For example, at $k = K$, $\mathbb{E}[\eta_K] = \mathcal{O}((G^2 + \sigma^2)/\beta_K T_K) = \mathcal{O}(\varepsilon)$.

Now taking expectation on both sides of (50) and using linearity of expectation and (51) we get that

$$\mathbb{E}[\Psi_\lambda\left(x_K, x_K'\right)] - \Psi_\lambda(x, x') \leq \frac{4(\|x_0 - x\|^2 + (3/2)\|x_0 - x'\|^2 + 2\tilde{D})}{\lambda K(K+1)} + \frac{\sum_{k=1}^K 2k\,\eta_k}{K(K+1)} \quad (53)$$

Next setting $x' = x = x^* \in \mathcal{X} \subseteq \mathcal{X}'$ and using Lemma 2 and (5) we get that

$$\mathbb{E}[f\left(x_K\right)] - f(x^*) \leq \frac{10\|x_0 - x^*\|^2 + 8\tilde{D}}{\lambda K(K+1)} + \frac{\sum_{k=1}^K 2k\,\eta_k}{K(K+1)} + G^2\frac{\lambda}{2} \quad (54)$$

*Aside:* Note that, in the final guarantee, when we set $\lambda = \varepsilon/G^2$, the third term, which is the error from the Moreau smoothing becomes $\varepsilon/2$. Additionally, when $K = \mathcal{O}(1/\varepsilon)$, first term above is $\mathcal{O}(\varepsilon)$. Further, when we set $T_k = \Theta(\varepsilon\,k^2) = \mathcal{O}(1/\varepsilon)$, we get $1/\beta_k = \mathcal{O}(1/\varepsilon\,k)$ and $\mathbb{E}[\eta_k] = \mathcal{O}(1/k)$ so that the second term is also $\mathcal{O}(\varepsilon)$. $\qquad\square$

Next using the above result we derive the guarantees for MOPES (Algorithm 1) and MOLES (Algorithm 2) as corollaries of Theorem 5.

### C.2.1 Proof of Theorem 1

*Proof.* Notice that exact projection on Line 1.5 of Algorithm 1 is equivalent to choosing $\eta_k = 0$ in Algorithm 4. Then setting, $\eta_k = 0$, and $\lambda = \varepsilon/G^2$ in Theorem 5 we get

$$\mathbb{E}[f\left(x_K\right)] - f(x^*) \leq \frac{G^2(10\|x_0 - x^*\|^2 + 8\tilde{D})}{\varepsilon K(K+1)} + 0 + \frac{\varepsilon}{2} \quad (55)$$

Now, by using the given choices: $\tilde{D} = c\|x_0 - x^*\|^2$ and $K = \lceil\frac{2\sqrt{10+8c}\,G\|x_0 - x^*\|}{\varepsilon}\rceil$, we get

$$\mathbb{E}[f\left(x_K\right)] - f(x^*) \leq \varepsilon \quad (56)$$

Then the number of PO calls made by the algorithm is $K = \mathcal{O}(\frac{G\|x_0 - x^*\|}{\varepsilon})$ and the total number of SFO calls made subgradients made is

$$\sum_{k=1}^K T_k \leq \sum_{k=1}^K \left(\frac{(4G^2 + \sigma^2)\lambda^2 K k^2}{2\tilde{D}} + 1\right) = \frac{(4G^2 + \sigma^2)\varepsilon^2 K^2(K+1)(2K+1)}{12cG^4\|x_0 - x_\lambda^*\|^2} + K$$

$$= \mathcal{O}\left(\frac{(G^2 + \sigma^2)\|x_0 - x^*\|^2}{\varepsilon^2}\right), \quad (57)$$

where we used Line 1.3 and the given choices for $\lambda$, $K$, and $\tilde{D}$. $\qquad\square$

### C.2.2 Proof of Theorem 2

*Proof.* Notice that at step $k$ of Algorithm 2 choosing $\hat{T} = \lceil\frac{7KD_{\mathcal{X}}^2}{c'\tilde{D}}\rceil = \mathcal{O}(\frac{1}{\varepsilon})$ is equivalent to choosing $\eta_k = \frac{4c'\tilde{D}}{\lambda K k}$ in Algorithm 4 (see below). Therefore by setting, $\eta_k = \frac{4c'\tilde{D}}{\lambda K k} = \mathcal{O}(\frac{1}{k})$, and $\lambda = \varepsilon/G^2$ in Theorem 5 we get

$$\mathbb{E}[f\left(x_K\right)] - f(x^*) \leq \frac{G^2(10\|x_0 - x^*\|^2 + 8\tilde{D} + 8c'\tilde{D})}{\varepsilon K(K+1)} + \frac{\varepsilon}{2} \quad (58)$$

Now, by using the given choices: $\tilde{D} = c\|x_0 - x^*\|^2$ and $K = \lceil\frac{2\sqrt{10+8c(1+c')}G\|x_0 - x^*\|}{\varepsilon}\rceil$, in the (58) we get

$$\mathbb{E}[f\left(x_K\right)] - f(x^*) \leq \varepsilon \quad (59)$$

Then using the similar arguments as in proof of Theorem 2, we can show that $K = \mathcal{O}(\frac{G\|x_0 - x^*\|}{\varepsilon})$ and the total number of SFO calls made is $\sum_{k=1}^K T_k = \mathcal{O}(\frac{(G^2+\sigma^2)\|x_0 - x^*\|^2}{\varepsilon^2})$.

Finally we calculate the total number of LMO calls made. At outer step $k$ of Algorithm 2, after using $\hat{T} = \lceil\frac{7KD_{\mathcal{X}}^2}{c'\tilde{D}}\rceil$ LMO calls in the `Approx-Proj` procedure, we can find a feasible point $z_k$ whose

Wolfe duality gap (38) is at most $\lceil \frac{6\beta_k D_{\mathcal{X}}^2}{\hat{T}} \rceil \leq \frac{4c'\tilde{D}}{\lambda K k} = \eta_k$ if we use CndG procedure [56, Theorem 2.2(c)] or $\lceil \frac{7\beta_k D_{\mathcal{X}}^2}{\hat{T}} \rceil \leq \frac{4c'\tilde{D}}{\lambda K k} = \eta_k$ if we use the standard Frank-Wolfe algorithm [45, Theorem 2]. Therefore the total number of linear minimization oracle calls made by the algorithm is

$$K\hat{T} = \frac{7K^2 D_{\mathcal{X}}^2}{cc'\|x_0 - x^*\|^2} + K = \mathcal{O}\Big(\frac{G^2 D_{\mathcal{X}}^2}{\varepsilon^2}\Big), \tag{60}$$

where we used Line 1.3 and the given choices for $K$ and $\tilde{D}$. $\qquad\square$

### C.2.3 Proof of Proposition 3: Analysis of `Prox-Slide` (Line 4.10) procedure

*Proof.* We analyze the `Prox-Slide` procedure for a fixed $k$, therefore we drop $k$ from $\phi_k$, $u_{k,t}$, $\widehat{g}_{k,t}$, and $\delta_{k,t}$, which are denoted here with $\phi$, $u_t$, $\widehat{g}_t$, and $\delta_t$. `Prox-Slide` has the following update steps.

$$\theta_t = \frac{2(t+1)}{t(t+3)}, \quad \widehat{g}_{t-1} = \text{SFO}\,(u_{t-1}) \tag{61}$$

$$\widehat{u}_t = u_{t-1} - \frac{1}{(1+t/2)\beta} \cdot (\widehat{g}_{t-1} + \beta(u_{t-1} - (u' - g/\beta))) \tag{62}$$

$$u_t = \widehat{u}_t \cdot \min(1, 2R/\|\widehat{u}_t\|) \tag{63}$$

$$\widetilde{u}_t = (1 - \theta_t)\widetilde{u}_{t-1} + \theta_t u_t \tag{64}$$

By convexity and $G$-Lipschitzness of $f$ in $\mathcal{X}'$, for any $u \in \mathcal{X}'$, we get

$$
\begin{aligned}
f(u_{t+1}) - f(u) &= f(u_{t+1}) - f(u_t) + f(u_t) - f(u) \\
&\leq \langle g_{t+1}, u_{t+1} - u_t \rangle + \langle g_t, u_t - u \rangle \\
&= \langle \widehat{g}_t, u_{t+1} - u_t \rangle + \langle g_{t+1} - g_t - \delta_t, u_{t+1} - u_t \rangle + \langle \widehat{g}_t, u_t - u \rangle - \langle \delta_t, u_t - u \rangle \\
&\leq \langle \widehat{g}_t, u_{t+1} - u \rangle + (2G + \|\delta_t\|)\|u_{t+1} - u_t\| + \langle \delta_t, u - u_t \rangle,
\end{aligned} \tag{65}
$$

where we used the fact that $\delta_t = \widehat{g}_t - g_t$. Notice that $u_t = \widehat{u}_t \cdot \min(1, 2R/\|\widehat{u}_t\|)$ is the projection of $\widehat{u}_t$ onto $\mathcal{X}' = B(0, 2R)$. Therefore, using Line 4.14, we can re-write `Prox-Slide` update as

$$
\begin{aligned}
u_{t+1} &= \underset{u \in \mathcal{X}'}{\text{argmin}}\, \frac{(t+3)\beta}{4}\|u - \widehat{u}_t\|^2 \\
&= \underset{u \in \mathcal{X}'}{\text{argmin}}\, \frac{(t+3)\beta}{4}\left\| u - \left(u_{t-1} - \frac{1}{(1+((t+1)/2)\beta)} \cdot (\widehat{g}_{t-1} + \beta(u_{t-1} - (u_0 - g/\beta)))\right)\right\|^2 \\
&= \underset{u \in \mathcal{X}'}{\text{argmin}}\, \left[F_t(u) := \langle g, u \rangle + \langle \widehat{g}_t, u \rangle + \frac{(t+1)\beta}{4}\|u - u_t\|^2 + \frac{\beta}{2}\|u - u_0\|^2\right]
\end{aligned} \tag{66}
$$

By $\beta(t+3)/2$-strong convexity of the quadratic update objective $F_t(u)$ and the optimality of $u_{t+1} \in \text{argmin}_{u \in \mathcal{X}'} F_t(u)$, we get that for any $u \in \mathcal{X}'$

$$\frac{\beta(t+3)}{4}\|u_{t+1} - u\|^2 + F_t(u_{t+1}) \leq F_t(u) \tag{67}$$

We want to provide a lower bound on $\phi(u)$ 44 which is defined as follows, when using the private notation of the `Prox-Slide` procedure by setting $u = x'$, $u' = z_{k-1}$, $g = \nabla_{k,x'}$, $\beta = \beta_k$.

$$\phi(u) := f(u) + \langle g, u \rangle + \frac{\beta}{2}\|u - u_0\|^2. \tag{68}$$

Now adding together (65) and (67) and using the definitions of $F_t$ and $\phi$ we get

$$
\begin{aligned}
\phi(u_{t+1}) - \phi(u) &\leq \frac{\beta}{2}\Big(\frac{t+1}{2}\|u_t - u\|^2 - \frac{t+3}{2}\|u_{t+1} - u\|^2\Big) + \\
&\quad (2G + \|\delta_t\|)\|u_{t+1} - u_t\| - \frac{\beta(t+1)}{4}\|u_{t+1} - u_t\|^2 + \langle \delta_t, u - u_t \rangle \\
&\leq \frac{\beta}{2}\Big(\frac{t+1}{2}\|u_t - u\|^2 - \frac{t+3}{2}\|u_{t+1} - u\|^2\Big) + \frac{(2G + \|\delta_t\|)^2}{\beta(t+1)} + \langle \delta_t, u - u_t \rangle
\end{aligned}
$$
$$\tag{69}$$

where the second inequality follows from $ax - bx^2/2 \le a^2/2b$. Now multiplying the above inequality is by $2(t+2)/T(T+3)$ and then summing over $t = \{0, \ldots, T-1\}$, we get

$$\sum_{t=0}^{T-1} \frac{2(t+2)}{T(T+3)}(\phi(u_{t+1}) - \phi(u)) \le \frac{\beta}{2}\frac{2}{T(T+3)}(\|u_0 - u\|^2 - \frac{(T+1)(T+2)}{2}\|u_T - u\|^2)+$$

$$\frac{2}{T(T+3)}\frac{2\sum_{t=0}^{T-1}(2G+\|\delta_t\|)^2}{\beta} + \sum_{t=0}^{T-1}\frac{2(t+2)}{T(T+3)}\langle \delta_t, u - u_t \rangle$$

$$\phi(\widetilde{u}_T) - \phi(u) \le \tag{70}$$

where the last inequality uses Jensen's inequality and $\widetilde{u}_T = \sum_{t=1}^{T} \frac{2(t+1)}{T(T+3)} u_t$, last of which follows from Lines 4.13 and 4.16 as follows

$$\widetilde{u}_T = (1 - \theta_T)\widetilde{u}_{T-1} + \theta_T u_T$$
$$= \frac{(T-1)(T+2)}{T(T+3)}((1 - \theta_{T-1})\widetilde{u}_{T-2} + \theta_{T-1}u_{T-1}) + \frac{2(T+1)}{T(T+3)}u_T$$
$$= \frac{(T-2)(T+1)}{T(T+3)}\widetilde{u}_{T-2} + \frac{2(T)}{T(T+3)}u_{T-1} + \frac{2(T+1)}{T(T+3)}u_T$$
$$\vdots$$
$$= \sum_{t=1}^{T}\frac{2(t+1)}{T(T+3)}u_t \tag{71}$$

Finally we get the desired result by setting $\phi = \phi_k$, $\beta = \beta_k$, $T = T_k$, $u_0 = \widetilde{z}'_{k-1}$, $u = x'$, $u_t = u_{k,t}$, $\widetilde{u}_T = \widetilde{z}'_k$, and $u_T = z_k$ we get the desired inequality

$$\phi_k(\widetilde{z}'_k) - \phi_k(x') \le \frac{2}{T_k(T_k+3)}\frac{\beta_k}{2}\|z'_{k-1} - x'\|^2 - \frac{(T_k+1)(T_k+2)}{T_k(T_k+3)}\frac{\beta_k}{2}\|z'_k - x'\|^2+$$

$$\frac{4\sum_{t=0}^{T_k-1}(2G+\|\delta_{k,t}\|)^2}{\beta_k T_k(T_k+3)} + \sum_{t=0}^{T_k-1}\frac{2(t+2)}{T_k(T_k+3)}\langle \delta_{k,t}, x' - u_{k,t} \rangle \tag{72}$$

$$\square$$

### C.3 Proof of Lemma 1

We re-write $f_\lambda(x)$ as minimum value of a $\frac{1}{\lambda}$-strong convex function $\phi_{\lambda,x}$ as follows

$$f_\lambda(x) = \min_{x' \in \mathcal{X}'}\left[\phi_{\lambda,x}(x') := f(x') + \frac{1}{2\lambda}\|x - x'\|^2\right]. \tag{73}$$

Note that $\phi_{\lambda,x}(\cdot)$ is a $(1/\lambda)$-strongly convex function as $f$ is convex and $(1/\lambda)\|\cdot - x\|^2$ is strongly convex, and $f_\lambda(x) = \min_{x' \in \mathcal{X}'} \phi_{\lambda,x}(x')$.
(a) The existence and uniqueness of $\hat{x}_\lambda(x) \in \mathcal{X}'$ follows from the strong convexity of $\phi_{\lambda,x}(\cdot)$ and the fact that $f$ is a proper convex function. Then $f(\hat{x}_\lambda(x)) \le \phi_{\lambda,x}(\hat{x}_\lambda(x)) = \min_{x' \in \mathcal{X}'} \phi_{\lambda,x}(x') = f_\lambda(x) \le \phi_{\lambda,x}(x) = f(x)$.
(b) Let $g_x := (x - \hat{x}_\lambda(x))/\lambda$ for any $x \in \mathbb{R}^d$. By $(1/\lambda)$-strong convexity of $\phi_{\lambda,x}(x')$ and $\hat{x}_\lambda(x) = \text{argmin}_{x' \in \mathcal{X}'} \phi_{\lambda,x}(x')$, we have, for any $x' \in \mathcal{X}'$, that

$$\phi_{\lambda,x}(x') \ge \phi_{\lambda,x}(\hat{x}_\lambda(x)) + \|x' - \hat{x}_\lambda(x)\|^2/2\lambda$$
$$\iff f(x') + \|x' - x\|^2/2\lambda \ge f(\hat{x}_\lambda(x)) + \|x' - \hat{x}_\lambda(x)\|^2/2\lambda + \|x' - \hat{x}_\lambda(x)\|^2/2\lambda$$
$$\iff f(x') \ge f(\hat{x}_\lambda(x)) + \langle g_x, x' - \hat{x}_\lambda(x) \rangle \tag{74}$$

Using this, for any $x, y \in \mathbb{R}^d$ we get

$$f_\lambda(y) - f_\lambda(x) = f(\hat{x}_\lambda(y)) - f(\hat{x}_\lambda(x)) + (\|\hat{x}_\lambda(y) - y\|^2 - \|\hat{x}_\lambda(x) - x\|^2)/2\lambda$$
$$\ge \langle g_x, \hat{x}_\lambda(y) - \hat{x}_\lambda(x) \rangle + \lambda/2(\|g_y\|^2 - \|g_x\|^2) = \langle g_x, y - x \rangle + \lambda/2\|g_x - g_y\|^2 \tag{75}$$

By instantiating the above for $y \leftarrow x$, $x \leftarrow y$, we also get $f_\lambda(y) - f_\lambda(x) \leq \langle g_y, y - x \rangle - \lambda/2\|g_x - g_y\|^2$. Combining these two inequalities

$$0 \leq \lambda/2\|g_y - g_x\|^2 \leq f_\lambda(y) - f_\lambda(x) - \langle g_x, y - x \rangle \leq -\lambda/2\|g_y - g_x\|^2 + \langle g_y - g_x, y - x \rangle$$
$$\leq -\lambda/2\|g_y - g_x\|^2 + \|g_y - g_x\|\|y - x\|$$
$$\leq \|y - x\|^2/2\lambda \tag{76}$$

This implies that $\lim_{y \to x}(f_\lambda(y) - f_\lambda(x) - \langle g_x, y - x \rangle)/\|y - x\| = 0$. Thus $f_\lambda$ is Frechet differentiable with gradient $\nabla f_\lambda(x) = g_x = (x - \hat{x}_\lambda(x))/\lambda$. The above inequality also implies $f_\lambda$ is convex and $1/\lambda$-smooth.

(c) Let $x \in \mathcal{X}'$. Using $1/\lambda$-strong convexity of $\phi_{\lambda,x}$ and $\hat{x}_\lambda(x) \in \arg\min_{x' \in \mathcal{X}'} \phi_{\lambda,x}(x')$, and $G$-Lipschitzness of $f$ in $\mathcal{X}'$, we get

$$\|x - \hat{x}_\lambda(x)\|^2/2\lambda \leq \phi_{\lambda,x}(x) - \phi_{\lambda,x}(\hat{x}_\lambda(x)) = f(x) - f_\lambda(x)$$
$$= f(x) - f(\hat{x}_\lambda(x)) - \|x - \hat{x}_\lambda(x)\|^2/2\lambda$$
$$\leq G\|\hat{x}_\lambda(x) - x\| - \|x - \hat{x}_\lambda(x)\|^2/2\lambda \leq G^2\lambda/2 . \quad \square$$

# D    Additional details for the experiments in Section 5

For all the experiments we randomly and uniformly sample a point $x_0$ from the surface of the nuclear norm ball of radius $r$. For all the figures where we plot the estimated sub-optimality gap: $f(x_k) - \hat{f}^*$, where $\hat{f}^*$ is the estimated minimum function value calculated by running the PGD method for a large number of iterations. We plot the mean (standard error is negligible) of the sub-optimality gap over 10 runs using 10 different initial points $x_0$'s (same 10 initial points for all algorithms).

For experiments in Figures 1 and 2, we use a subset of the Imagewoof 2.0 dataset [43], which in itself is a subset of the Imagenet dataset [24]. The training data, contains $n = 400$ samples $\{(A_i, y_i)\}_{i=1}^n$ where $A_i$ is a $224 \times 224$ grayscale image of one of the two types of dogs (classes n02087394 and n02115641 in Imagenet dataset) labeled using $y_i \in \{0, 1\}$. Note that the effective dimension is $d = 224 \times 224 = 50176$). These grayscale images are generated from the raw 8-bit RGB Imagewoof images using the Pillow python image-processing library [21], by $(i)$ resizing to $256 \times 256$ pixels: `resize(256,256)`, $(ii)$ cropping to the central $224 \times 224$ pixels: `crop(16,16,240,240)`, $(iii)$ converting to the grayscale: `convert(mode='L')`, and $(iv)$ normalizing by 255.0 so that the pixel values lie in range $[0, 1]$. For incorporating bias scalar into the SVM model we also zero-pad the training images with an additional column and row of zeros to the right and the bottom of the image array $A_i$. We use $r = 0.1$ as nuclear norm ball radius of $\mathcal{X}$, thus $D_\mathcal{X} = 0.2$. We have access to a deterministic FO.

In Figure 1, we use a Lipschitz constant of $G = 50$. For MOPES we set $c = 40$ and $\varepsilon = 5.0$, and for PGD we use two stepsize schemes: $(i)$ fixed stepsize $D_\mathcal{X}/(G\sqrt{K})$ with $K = 10^3$ and $(ii)$ diminishing stepsize $D_\mathcal{X}/(G\sqrt{k})$ with $K = 10^3$.

In Figure 2, we use a Lipschitz constant of $G = 50$. For MOLES we set $c = 40$, $c' = 1$ and $\varepsilon = 5.0$. For FW-PGD we use two stepsize schemes: $(i)$ fixed stepsize $D_\mathcal{X}/(G\sqrt{K})$ with $K = 10^3$ and $(ii)$ diminishing stepsize $D_\mathcal{X}/(G\sqrt{k})$ with $K = 10^3$. Both of these stepsize schemes use a projection tolerance of $\eta_k G^2/2$. For RandFW we use the standard parameter choices as given in [54, Theorem 5] with $K = 150$.

In practice, in the deterministic setup with FO, at outer-step $k$, we can use the following criterion for stopping the `Prox-Slide` (Line 1.8) procedure early at some $t \geq \hat{T}_{k-1}$ (defined recursively below with $\hat{T}_0 = 1$) and $t \leq T_k$. Let $\phi_k(x') := f(x') + \langle \nabla_{k,x'}, x' \rangle + \frac{\beta_k}{2}\|x' - z'_{k-1}\|^2$ and $\tilde{g}_t \in \partial f(\tilde{u}_t)$. Now if

$$\max_{x' \in \mathcal{X}} \langle \tilde{g}_t + \nabla_{k,x'}, \tilde{u}_t - x' \rangle - \frac{(t+1)(t+2)}{t(t+3)}\beta_k \langle u_t - z'_{k-1}, x' \rangle$$
$$\leq \frac{8(4G^2 + \sigma^2)}{\beta_k(T_k + 3)} - \frac{\beta_k}{2}\|\tilde{u}_t - z'_{k-1}\|^2 + \frac{(t+1)(t+2)}{t(t+3)}\frac{\beta_k}{2}(\|z'_{k-1}\|^2 - \|u_t\|^2) \tag{77}$$

then we stop the procedure, set $\widehat{T}_k = t$ and return $(u_t, \widetilde{u}_t)$. This implies that for $(z'_k, \widetilde{z}'_k) = (u_t, \widetilde{u}_t)$

$$
\phi_k(\widetilde{u}_t) - \phi_k(x') \le \frac{2}{\widehat{T}_k(\widehat{T}_k + 3)} \frac{\beta_k}{2} \|z'_{k-1} - x'\|^2 - \frac{(\widehat{T}_k + 1)(\widehat{T}_k + 2)}{\widehat{T}_k(\widehat{T}_k + 3)} \frac{\beta_k}{2} \|u_t - x'\|^2 +
$$
$$
\frac{4 \sum_{t=0}^{T_k - 1} (2G)^2}{\beta_k T_k (T_k + 3)} \tag{78}
$$

for all $x' \in \mathcal{X}$. Now the only change we need to make in the analysis of Theorem 5 is the change of the potential (40) to

$$
\Phi_k := k(k+1)(\Psi_\lambda(x_k, x'_k) - \Psi_\lambda(x, x')) + \frac{4}{\lambda}(\|z_k - x\|^2 + \frac{(\widehat{T}_{k+1} + 1)(\widehat{T}_{k+1} + 2)}{\widehat{T}_{k+1}(\widehat{T}_{k+1} + 3)} \|z'_k - x'\|^2) \tag{79}
$$

The LHS of (77) is an linear optimization problem whose solution can be easily found as

$$
\mathrm{LMO}\left( \widetilde{g}_t + \nabla_{k,x'} + \frac{(t+1)(t+2)}{t(t+3)} \beta_k(u_t - z'_{k-1}) \right)
$$
$$
= \lim_{\alpha \to \infty} \mathrm{PO}\left( -\alpha\left( \widetilde{g}_t + \nabla_{k,x'} + \frac{(t+1)(t+2)}{t(t+3)} \beta_k(u_t - z'_{k-1}) \right) \right). \tag{80}
$$

We also use a slightly modified $T_k = \left\lceil \frac{2G^2 \lambda^2 K k^2}{2\tilde{D}} \right\rceil$ for our experiments, since the deterministic FO we use, ensures this choice gets the same guarantees as given in our theorems. Also, in our implementation we do not explicitly project $z_k$ onto $\mathcal{X}'$, as in practice this does not seem needed.

In practice, we can eliminate the need for selecting $\varepsilon$ of MOPES by employing $\varepsilon$-doubling trick with warm restarts, which can increase the worse-case iteration complexity by a factor of at most 2 but oftentimes will accelerate the convergence [80, Algorithm 6].

# E  Additional details for applications

We refer to [73] for some more nonsmooth problems which can be solved using an LMO. In the following subsections, we compare the analytical complexities for solving some of the applications mentioned in Section 4, using different algorithms.

## E.1  $\ell_1$ norm constrained SVM

For simplicity, we work with the vector version of the matrix problems and replace nuclear norm constraint with the $\ell_1$ norm constraint. The standard $\ell_1$ norm constrained soft-margin SVM can be formulated as the the following optimization problem:

$$
\min_{x \in \mathbb{R}^d} f(x) = \frac{1}{n} \sum_{i=1}^n [f_i(x) = \max(0, 1 - \langle x, a_i \rangle)]
$$
$$
\text{subject to} \quad \|x\|_1 \le \lambda \tag{81}
$$

where $a_i \in \mathbb{R}^d$ captures the $d$-dimensional feature vector multiplied by a binary class value in $\{-1, 1\}$ and $\mathcal{X} = \{x \mid \|x\|_1 \le 1\}$ is the constraint set. We do not include any explicit bias term above, because it can always be incorporated into the model by augmenting $a_i$ with a constant dimension. We assume that $n$ is large and therefore we only have access to minibatched stochastic subgradients obtained through minibatching $d$ ($b = o(n)$) uniformly sampled (with replacement) training samples. We assume that $f$ is $G_p$-Lipschitz continuous and the variance of any stochastic subgradient is upperbounded by $\sigma_p^2$, both calculated in $\ell_p$ norm $\| \cdot \|_p$, for $p = 1, 2$. We define $q := (1 - 1/p)^{-1} \in \{\infty, 2\}$. Then

$$
G_p = \max_{\|x\|_1 \le \lambda} \|\frac{1}{n} \sum_{i=1}^n \mathbb{I}\{\langle x, a_i \rangle < 1\} a_i\|_q, \text{ and}
$$

$$
\sigma_p^2 = \max_{\|x\|_1 \le \lambda} \mathbb{E}_{\{I_j\}_{j=1}^b} \|\frac{1}{n} \sum_{i=1}^n \mathbb{I}\{\langle x, a_i \rangle < 1\} a_i - \frac{1}{b} \sum_{j=1}^b \mathbb{I}\{\langle x, a_{I_j} \rangle < 1\} a_{I_j}\|_q^2 \tag{82}
$$

| PO based methods (using $\ell_p$ norm) | | |
| --- | --- | --- |
| **Nonsmooth methods ($p = 2$)** | **PO**: $\mathcal{O}(d \ln d)$ | **SFO**: $\mathcal{O}(d + n)$ |
| Our MOPES ($p = 2$) [Theorem 1] | $\mathcal{O}\big(\frac{G_2}{\varepsilon}\lambda\big)$ | $\mathcal{O}\big(\frac{G_2^2+\sigma_2^2}{\varepsilon^2}\lambda^2\big)$ |
| PGD ($p = 2$) | $\mathcal{O}\big(\frac{G_2^2}{\varepsilon^2}\lambda^2\big)$ | $\mathcal{O}\big(\frac{G_2^2+\sigma_2^2}{\varepsilon^2}\lambda^2\big)$ |
| Randomized smoothing ($p = 2$) [27] | $\mathcal{O}\big(d^{1/4}\frac{G_2}{\varepsilon}\lambda\big)$ | $\mathcal{O}\big(\frac{G_2^2+\sigma_2^2}{\varepsilon^2}\lambda^2\big)$ |
| **Nonsmooth methods ($p = 1$)** | **MO**: $\mathcal{O}(d)$ | **SFO**: $\mathcal{O}(d + n)$ |
| Mirror descent ($p = 1$) [64] | $\mathcal{O}\big(\ln(d+1)\frac{G_1^2}{\varepsilon^2}\lambda^2\big)$ | $\mathcal{O}\big(\ln(d+1)\frac{G_1^2+\sigma_1^2}{\varepsilon^2}\lambda^2\big)$ |
| Randomized smoothing ($p = 1$) [27] | $\mathcal{O}\big(\sqrt{d\ln(d+1)}\frac{G_1}{\varepsilon}\lambda\big)$ | $\mathcal{O}\big(\ln(d+1)\frac{G_1^2+\sigma_1^2}{\varepsilon^2}\lambda^2\big)$ |
| **Minimax methods**: $\mathcal{O}(n)$ extra memory | **PO+MO**: $\mathcal{O}(d \ln d + n)$ | **SFO**: $\mathcal{O}(d + n)$ |
| Variance reduced Mirror-Prox ($p = 1$)[16] | $\mathcal{O}\left( \frac{dn}{d+n} + \frac{L_{12}}{\varepsilon}\sqrt{\frac{dn}{d+n}}(\lambda\sqrt{n\ln d})\right)$ | |

Table 2: Projection: Comparison of PO/MO and SFO calls complexities (PO-CC and SFO-CC) of various methods for $d$-dimensional $\ell_1$ norm constrained SVM with $n$ training samples. SFO uses a batchsize of $b = o(n)$. Our MOPES outperforms other nonsmooth methods in PO-CC/MO-CC while still maintaining $\mathcal{O}(1/\varepsilon^2)$ SFO-CC. Complexities of methods based on smooth minimax reformulation adversely scale with $n$ or $d$.

**PO:** First we study the case of PO (or MO: Mirror descent step oracle) in the high-dimensional ($\mathrm{poly}(G_p, \sigma_p, \lambda, 1/\varepsilon) \ll d$) and large-scale ($1 \ll \mathrm{poly}(n)$) regime. In Table 2 we provide the PO-CC and SFO-CC of MOPES ($p = 1$, Algorithm 1) and competing nonsmooth methods: PGD ($p = 2$) [34, 59], Mirror descent ($p = 1$) [64], Randomized smoothing ($p = 1$ or $p = 2$) [27]. The $p$ value in brackets marks which $\ell_p$ norm the method uses. By definition $G_1 \leq G_2 \leq \sqrt{d}G_1$ and $\sigma_1 \leq \sigma_2 \leq \sqrt{d}\sigma_1$. Therefore, in this high-dimensional and large-scale regime, and when $G_2 = o(\sqrt{d}G_1)$ and $\sigma_2 = o(\sqrt{d}\sigma_1)$, MOPES has a more efficient PO-CC than other competing nonsmooth first-order methods, while still maintaining $\mathcal{O}(1/\varepsilon^2)$ SFO-CC. Note that PO has a computational complexity of $O(d \log d)$ because it involves sorting [26], MO has a computational complexity of $\mathcal{O}(d)$, and SFO has a computational complexity of $\mathcal{O}(b(d + n))$ because it involves sampling $b$ vectors from a set of $n$ $d$-dimensional vectors. In practice, sorting could contribute to a significant part of the wall-clock time.

Many nonsmooth convex objectives in machine learning like the hinge loss here can be written as smooth convex-concave minimax objectives of the form

$$\min_{x \in \mathcal{X}} \frac{1}{n} \sum_{i=1}^{n} \max_{y_i \in \mathcal{Y}_i} g_i(x, y_i) \tag{83}$$

where $g_i(x, \cdot)$ is concave and $g_i$ is $L$-smooth for all $i \in [n]$. However, the iteration/projection complexities of even the best variance reduced algorithms could have a dependence on the number $n$ of additionally introduced dual variables $\{y_i\}_{i=1}^n$ [71, 16]. Therefore in the regime when comparatively $n$ is large and $\varepsilon$ is moderate ($\mathrm{poly}(1/\varepsilon) \ll n$), it is more efficient to optimize the original stochastic nonsmooth formulation than the smooth minimax reformulation.

Concretely, the soft-margin SVM problem with a hinge loss, can be reformulated as a saddle point problem of the following form

$$\min_{\|x\|_1 \leq 1} \max_{y \in [0,1]^n} \frac{1}{n}(y^T \mathbf{1} - y^T Ax) . \tag{84}$$

This smooth saddle point problem is an $\ell_1$-$\ell_2$ matrix game (ignoring possibility of $\ell_\infty$ optimization due to limited literature) which is $L_{12}$-smooth, where

$$L_{12} = \max_{\|x\|_1 \leq 1} \max_{\|y\|_2 \leq 1} \frac{1}{n} y^T Ax = \frac{1}{n} \max_{i=1,\ldots,n} \|a_i\|_2 . \tag{85}$$

Note that the primal (in $\ell_1$ norm) and dual (in $\ell_2$ norm) space diameters are $D_{\mathcal{X}} = \mathcal{O}(\lambda)$ and $D_{\mathcal{Y}} = \mathcal{O}(\sqrt{n})$ respectively.

Next we derive the MO and SFO calls complexities (MO-CC and SFO-CC) of the variance reduced Mirror-prox method [16]. For any stepsize $\alpha \leq \frac{\varepsilon}{D_{\mathcal{X}} D_{\mathcal{Y}} \sqrt{\ln d}}$, this algorithm runs for $K = \mathcal{O}(\frac{\alpha D_{\mathcal{X}} D_{\mathcal{Y}} \sqrt{\ln d}}{\varepsilon})$ outer iterations, each of which uses $T = 1 + \frac{L_{12}^2}{\alpha^2}$ SFO calls and one FO call, and $T + 1$ primal and dual MO calls. Computational complexity of

- Primal MO is $O(d \log d)$ since it involves sorting,
- Dual MO is $O(n)$ since it involves normalization of each dual dimension,
- FO is $\mathcal{O}(dn)$ since it involves $d \times n$-matrix vector products, and,
- SFO is $\mathcal{O}(d+n)$ because it involves sampling from two set of $n$ and $d$ ($d$ and $n$-dimensional, respectively) vectors.

We assume that the algorithm uses $\widetilde{T} = 1 + \frac{L_{12}^2}{\alpha^2} + \frac{dn}{d+n} = \mathcal{O}(\frac{L_{12}^2}{\alpha^2} + \frac{dn}{d+n})$ SFO calls per outer iteration, because computationally it is equivalent to $T = 1 + \frac{L_{12}^2}{\alpha^2}$ SFO calls and one FO call per outer iteration. Using the suggested stepsize $\alpha = \max(\frac{\varepsilon}{D_{\mathcal{X}} D_{\mathcal{Y}} \sqrt{\ln d}}, L_{12} \sqrt{\frac{d+n}{dn}})$, we get that

$$[\text{MO-CC} = \mathcal{O}(K \cdot T)] = \mathcal{O}\left(\frac{dn}{d+n} + \frac{L_{12}}{\varepsilon} \sqrt{\frac{dn}{d+n}} (\lambda \sqrt{n \ln d})\right) = [K \cdot \widetilde{T} = \text{SFO-CC}]. \quad (86)$$

In very high dimensional regime ($n \ll d$) or very large-scale regime ($d \ll n$), MO-CC of this smooth minimax formulation is $\mathcal{O}(d)$ or $\mathcal{O}(n)$ larger than PO-CC for MOPES. Further more the former method uses extra $\Theta(n)$ extra space for storing the dual variables.

**LMO:** Next we study the case of LMO in the high-dimensional ($\text{poly}(G_p, \sigma_p, \lambda, 1/\varepsilon) \ll d$) and large-scale ($1 \ll \text{poly}(n)$) regime. In Table 3 we provide the LMO and SFO calls complexities of MOLES ($p = 1$, Algorithm 1) and competing nonsmooth methods: FW-PGD ($p = 2$)—projection approximated with Frank-Wolfe method (Appendix B.2), and Randomized Frank-Wolfe method ($p = 1$ or $p = 2$) [54]. The $p$ value in brackets marks which $\ell_p$ norm the method uses. By definition $G_1 \leq G_2 \leq \sqrt{d} G_1$ and $\sigma_1 \leq \sigma_2 \leq \sqrt{d} \sigma_1$. Therefore, in this high-dimensional and large-scale regime, MOLES has a more efficient dimension-free LMO-CC $\mathcal{O}(G_2^2 \lambda^2 / \varepsilon^2)$ than other competing nonsmooth first-order methods, while still maintaining optimal $\mathcal{O}(1/\varepsilon^2)$ SFO-CC. Note that LMO has a computational complexity of $O(d)$ because it uses just one pass over a $d$-dimensional vector.

A competing method based on the smooth minimax reformulation is SP+VR-MP which combines ideas from Semi-Proximal [41] and Variance reduced [16] Mirror-Prox methods. Here SP+VR-MP uses the variance reduced Mirror-prox method [16] in the $\ell_2$-$\ell_2$ setting to optimize (84) and then approximates the projection steps with Frank-Wolfe (FW) method. This is an $L_{22}$-smooth minimax problem with

$$L_{22} = \max_{\|x\|_2 \leq 1} \max_{\|y\|_2 \leq 1} \frac{1}{n} y^T A x = \frac{1}{n} \|A\|_2. \quad (87)$$

where $\|A\|_2$ is the spectral norm of the matrix $A$, but the algorithm we are discussing will depend on

$$\widetilde{L}_{22} = \frac{1}{n} \|A\|_F. \quad (88)$$

where $\|A\|_F$ is the Frobenius norm of the matrix $A$. Note that $\frac{1}{\sqrt{\min(n,d)}} \|A\|_F \leq \|A\|_2 \leq \|A\|_F$. The primal and dual space diameters are again $D_{\mathcal{X}} = \mathcal{O}(\lambda)$ and $D_{\mathcal{Y}} = \mathcal{O}(\sqrt{n})$, respectively.

For each of the projection steps, the Frank-Wolfe method solves an $(\alpha + 10 \frac{L_{22}^2}{\alpha})$-smooth convex optimization problem up to an error $\mathcal{O}(\varepsilon)$. Therefore each of these uses at most $\widehat{T} = \lceil (\alpha + 10 \frac{L_{22}^2}{\alpha}) \lambda^2 / \varepsilon \rceil$ LMO calls. Thus using similar arguments as the PO setting and using the suggested stepsize $\alpha = \max(\frac{\varepsilon}{D_{\mathcal{X}} D_{\mathcal{Y}}}, \widetilde{L}_{22} \sqrt{\frac{d+n}{dn}})$, $K = \mathcal{O}(\frac{\alpha D_{\mathcal{X}} D_{\mathcal{Y}}}{\varepsilon})$ outer iterations, $\widetilde{T} = 1 + \frac{\widetilde{L}_{22}^2}{\alpha^2} = \mathcal{O}(\frac{\widetilde{L}_{22}^2}{\alpha^2})$ SFO calls per outer iteration, and $\widetilde{T} = 1 + \frac{\widetilde{L}_{22}^2}{\alpha^2} + \frac{dn}{d+n} = \mathcal{O}(\frac{\widetilde{L}_{22}^2}{\alpha^2} + \frac{dn}{d+n})$ effective number of SFO calls per outer iteration, we get that

$$[\text{SFO-CC} = K \cdot \widetilde{T}] = \mathcal{O}\left(\frac{dn}{d+n} + \frac{\widetilde{L}_{22}}{\varepsilon} \sqrt{\frac{dn}{d+n}} (\lambda \sqrt{n})\right), \quad (89)$$

| LMO based methods (using $\ell_p$ norm) | | |
|---|---|---|
| **Nonsmooth methods ($p = 2$)** | **LMO**: $\mathcal{O}(d)$ | **SFO**: $\mathcal{O}(d+n)$ |
| MOLES ($p=2$) [Theorem 2] | $\mathcal{O}\big(\frac{G_2^2}{\varepsilon^2}\lambda^2\big)$ | $\mathcal{O}\big(\frac{G_2^2+\sigma_2^2}{\varepsilon^2}\lambda^2\big)$ |
| FW-PGD ($p=2$) [Theorem 4] | $\mathcal{O}\big(\frac{G_2^4+\sigma_2^4}{\varepsilon^4}\lambda^4\big)$ | $\mathcal{O}\big(\frac{G_2^2+\sigma_2^2}{\varepsilon^2}\lambda^2\big)$ |
| Rand. Frank-Wolfe ($p=2$) [54] | $\mathcal{O}\big(d^{1/2}\frac{G_2^2}{\varepsilon^2}\lambda^2\big)$ | $\mathcal{O}\big(\frac{G_2^4+\sigma_2^4}{\varepsilon^4}\lambda^4\big)$ |
| **Nonsmooth methods ($p = 1$)** | **LMO**: $\mathcal{O}(d)$ | **SFO**: $\mathcal{O}(d+n)$ |
| Rand. Frank-Wolfe ($p=1$) [54] | $\mathcal{O}\big(d\ln(d+1)\frac{G_1^2}{\varepsilon^2}\lambda^2\big)$ | $\mathcal{O}\big(\ln^2(d+1)\frac{G_1^4+\sigma_1^4}{\varepsilon^4}\lambda^2\big)$ |
| **Minimax methods**: $\mathcal{O}(n)$ extra memory | **LMO**: $\mathcal{O}(d)$ | **SFO**: $\mathcal{O}(d+n)$ |
| SP [41]+VR [16]-MP ($p=2$) | **SFO-CC** + $\mathcal{O}\Big(\frac{d\sqrt{n}\lambda}{d+n}+$ $\frac{\widetilde{L}_{22}\lambda^2}{\varepsilon}\big(\frac{dn}{d+n}\big)^{\frac{3}{2}}+\frac{\widetilde{L}_{22}^2\lambda^3\sqrt{n}}{\varepsilon^2}\big(\frac{dn}{d+n}\big)\Big)$ | $\mathcal{O}\Big(\frac{dn}{d+n}+$ $+\frac{\widetilde{L}_{22}}{\varepsilon}\sqrt{\frac{dn}{d+n}}(\lambda\sqrt{n})\Big)$ |

Table 3: Linear minimization oracle: LMO and SFO calls complexity (LMO-CC and SFO-CC) of various methods for $d$-dimensional $\ell_1$ norm constrained SVM with $n$ training samples. SFO uses a batchsize of $b = o(n)$. SP+VR-MP combines ideas from Semi-Proximal [41] and Variance reduced [16] Mirror-Prox methods. Our MOLES outperforms other nonsmooth methods in LMO-CC while still maintaining $\mathcal{O}(1/\varepsilon^2)$ SFO-CC. Complexities of method based on smooth minimax reformulation adversely scale with $n$ or $d$.

and

$$[\text{LMO-CC} = \mathcal{O}(K \cdot T) \cdot \widehat{T}]$$

$$= \mathcal{O}\Big(\Big[\frac{dn}{d+n} + \frac{\widetilde{L}_{22}}{\varepsilon}\sqrt{\frac{dn}{d+n}}(\lambda\sqrt{n})\Big] \cdot \Big[1 + (\alpha + \frac{L_{22}^2}{\alpha})\frac{D_{\mathcal{X}}^2}{\varepsilon}\Big]\Big)$$

$$= \mathcal{O}\Big(\Big[\frac{dn}{d+n} + \frac{\widetilde{L}_{22}}{\varepsilon}\sqrt{\frac{dn}{d+n}}(\lambda\sqrt{n})\Big] \cdot \Big[1 + \frac{\lambda}{\sqrt{n}} + \frac{\widetilde{L}_{22}\lambda^2}{\varepsilon}\sqrt{\frac{dn}{d+n}}\Big]\Big)$$

$$= \text{SFO-CC} + \mathcal{O}\Big(\frac{d\sqrt{n}\lambda}{d+n} + \frac{\widetilde{L}_{22}\lambda^2}{\varepsilon}\Big(\frac{dn}{d+n}\Big)^{\frac{3}{2}} + \frac{\widetilde{L}_{22}^2\lambda^3\sqrt{n}}{\varepsilon^2}\Big(\frac{dn}{d+n}\Big)\Big) \tag{90}$$

In very high dimensional regime ($n \ll d$) or very large-scale regime ($d \ll n$), LMO-CC of this smooth minimax reformulation is $\mathcal{O}(d)$ or $\mathcal{O}(n)$ larger than LMO-CC for MOLES. Further more the former method uses extra $\Theta(n)$ extra space for storing the dual variables.

Similar arguments hold for the nuclear norm constrained Matrix SVM [85], so that MOPES/MOLES outperforms other nonsmooth methods in some regime, where $n$ or $d$ is large and $\varepsilon$ is relatively moderate, and complexities of smooth minimax reformulation based methods scales adversely with $d$ and $n$. For this case, the gain in the actual wall-clock time would be even more stark than vector SVM due to the computation of SVD/largest eigenvalue, which is required for implementing PO/LMO.

### E.2 SVM with hard constraints

Soft-margin SVM could be provided with some hard constraints [70], so that the classifier is forced to always predict the correct labels for a subset (of size $k$) of important "gold" training examples. This problem can be formulated as a nonsmooth constrained optimization problem with a large number of linear constraints, as follows

$$\min_{x \in \mathbb{R}^d} \quad \frac{1}{n}\sum_{i=1}^{n}\max(0, 1 - \langle x, a_i \rangle)$$
$$\text{subject to,} \quad 1 \le \langle x, \widetilde{a}_j \rangle \,,\ \forall j = 1, \ldots, k$$
$$\|x\|_1 \le \lambda \tag{91}$$

We can solve this nonsmooth convex problem using first-order methods using projection onto hard constraints set: $\mathcal{X} = \{x \mid \|x\|_1 \le \lambda \text{ and } 1 \le \langle x, \widetilde{a}_j \rangle\,, \forall j = 1, \ldots, k\}$. This projection can be can be

implemented using linear programming methods, however it is computationally costly. Therefore PO-CC efficiency is critical, and just as in the case of SVM with $\ell_1$ norm constraint (Section E.1), our MOPES method achieves smallest $\mathcal{O}(1/\varepsilon)$ dimension-free PO-CC which is better than other competing methods. PO-CC and SFO-CC are the same as given in Table 2.

Note that, first-order methods using one projection [61] cannot be applied here, since they need the constraint set to be written in the functional form: $c(x) \leq 0$, such that $\rho \leq \|g\|$ for all $g \in \partial c(x)$, for some $\rho > 0$. This is not true for general set of linear constraints, where a pathological case can occur when two linear constraints have almost identical normal vectors.