[Reviews · NeurIPS 2020]

Review 1

Summary and Contributions: This paper considers the minimization of a nonsmooth Lipschitz continuous function over a convex and compact domain. The prototypical method for solving such problems is the projected subgradient method (PGD). At each iteration, PGD queries the subgradient oracle (FO) and the projection oracle (PO) for once. The iteration complexity of PGD for finding an eps-suboptimal solution is in the order of eps^{-2}, which is known to be optimal in terms of the FO calls. In this setting, the fundamental question that this paper asks is whether we can economize on the PO calls. The paper provides an affirmative answer to this question. Here is a summary of the contributions: - The paper introduces a new subgradient method (MOPES) that economizes the PO-calls. The method essentially solves a smooth approximation of the original problem based on the Moreau envelope by using a proximal accelerated gradient method, where the prox subproblems (strongly convex) are solved in an approximate sense. The resulting method has O(1/eps^2) FO complexity (which is optimal and already achieved by PGD), but it enjoys O(1/eps) PO complexity (in comparison to O(1/eps^2) for state-of-the-art). - The paper introduces a new projection-free subgradient method (MOLES). MOLES is a modification of MOPES where the projection oracle is replaced by an inexact projection oracle that can be implemented by the Frank-Wolfe method. MOLES has O(1/eps^2) linear minimization oracle (LMO) complexity and FO complexity. MOLES is the first method that achieves this (optimal) complexity. - The paper extends the proposed methods for the stochastic setting (unbiased estimator of the subgradient with bounded variance). ---- Update after the author feedback: I was concerned that the technical error in Lemma 1 is a fundamental flaw for the analysis, but I am happy that it turned out to be a lack of precision in the presentation. I adjusted my score to reflect the novelty and the significance that I underlined already in my original review, but please do not neglect to revise the writing.

Strengths: The novelty of the paper is its main strength. I am not aware of any prior work that investigates the fundamental question asked in this paper: Can we design subgradient methods that economize the calls to the projection oracles. I am also not aware of any projection-free method that achieves optimal rates for the nonsmooth problems. Projection-free methods have received an adequate level of attention from the ML community. A major part of this literature appeared in top ML conferences in the last decade. Along this line, I would expect this work to attract one attention too. Since there are not many alternatives for the projection-free subgradient methods in the literature, the paper might have some practical significance too. However, it is difficult to judge this aspect of the work due to the lack of empirical evidence.

Weaknesses: My primary concern is the correctness of the results. See Section Correctness for details. I can raise my score if the authors clarify my concern. The numerical experiments are weak. There is only a single experiment, on the low-rank SVM problem. The data used in this experiment is very small in size (d=64 dimensional problem), so it is not an interesting setting to be solved by using a projection-free method. There are many parameters involved in the algorithm and the experimental setup. The results are shown only for a particular choice. It is not clear how sensitive the proposed method’s performance to the choice of these parameters. Overall, there is not enough empirical evidence to comment on the practical significance of the proposed method. The quality of the presentation is weak, the mathematical writing of the paper makes it difficult to follow. (See Section Clarity).

Correctness: Lemma 1, introduced without the proof or any reference, plays an important role in the proof. This lemma presents standard properties of the Moreau envelope, and it is easy to find the proof for the case “Xbmm” is the Euclidean vector space. However, in the proposed approach, “Xbbm” is a Euclidean norm-ball. In this case, I do not think that property (c) in Lemma 1 is true. In particular, if you consider f as a constant function, then G=0, while hatxlambda(x) is the projection of x onto Xbbm, so this result does not hold. In the algorithm, it looks like “Xbbm” is the Euclidean norm-ball (step 1.13 is the projection). I tried to track “Xbbm” in the analysis. At line 557, it says “for simplicity we assume that … Xbbm is the whole vector space.” I do not see another remark in the rest of the paper why this assumption is -without loss of generality-.

Clarity: The paper requires polishing. The high-level claims of the paper are clear, but the mathematical writing is not. The authors drop mathematical details in the definitions and theorems, such as “for all x in X” etc, and this makes the reading cumbersome. The lack of these details can easily mislead the reader, I have personally found myself confused about the notation many times in my first read. You can find examples starting from Definitions 1 and 2 (Lipschitz continuity, strong convexity), where x and y are not defined, and similar issues exist in many parts of the paper. Most importantly, “Xbbm” is defined in Section 2 as a Euclidean norm ball around the origin with a fixed radius. In many other places, “Xbbm” is considered as a Euclidean vector space (e.g., Definition 3, and in many parts of the proof). This discrepancy also causes me to question the correctness of the results presented in the paper (see Section Correctness).

Relation to Prior Work: The paper presents the prior work in Section 1.1 under 5 sub-categories. The topic of “nonsmooth convex minimization” in general is too broad to cover exhaustively in one page. In my opinion, this section provides a good overview of this broad topic. The novelty of the proposed work (efficiency in terms of the PO-oracles, optimal LMO complexity of the projection-free version, the fundamental idea of smoothing and splitting behind the framework) is delivered in a clear way. The authors compare the guarantees of the proposed method with state of the art in Table 1.

Reproducibility: Yes

Additional Feedback: Some other minor comments: - Typo@line114: sest’s -> set’s - line 183: “PO calls are required in inner steps” -> is it “PO calls are NOT required in inner steps” ? - Typo@equation (22) - Typo@line667: “Similar to prosposition” -> proposition - Step 3.16 in prox-slide: What is “Xbbm” in step 3.15? If it is the Euclidean-norm ball, then step 3.16 is redundant. - Section D: Consider putting a reference or clarifying the procedure with numbered steps. Also, I did not understand why this section is titled “additional experimental details”.


Review 2

Summary and Contributions: This paper provides an improved algorithm for constrained minimization of non-smooth functions. The improvement is in the number of calls to the projection oracle, which may be vey expensive in general. For non-smooth optimization eps^(-2) iterations are required if no further structure is present — and this bound is very easily achieved via projected subgradient descent. T The algorithm presented here has the same iteration bound. However, under the very mild assumption that the function to minimize f is defined in a region slightly outside the constraint domain, it improves the number of calls to the projection oracle to eps^(-1). In addition to this, the authors also present an algorithm for the case where rather than a projection oracle, they have access to a linear optimization oracle for the domain. This algorithm is tight both in the number of iterations and the number of oracle calls. The main idea behind these algorithms is nice and clean. A first attempt would be to smoothen the function and solve using an accelerated first order method. This would not improve the convergence, since each iteration of the accelerated method would now require solving a nontrivial quadratic optimization problem, which adds an extra number of iterations that bring the iteration bound back to eps^(-2). However, this optimization step does not require projections, and this is where the saving comes from. ---- Update after the author feedback: The authors seem to have addressed all the reviewer concerns, and I maintain my score.

Strengths: The paper studies a fundamental problem in optimization, and the result looks quite surprising. It’s an important result that deserves acceptance.

Weaknesses: No major weaknesses.

Correctness: Everything checks out.

Clarity: The paper is well written and reads smoothly.

Relation to Prior Work: Yes

Reproducibility: Yes

Additional Feedback: Minor comments: line 114: “sest’s functional form” what does this mean? line 115: Wouldn’t it make more sense to have E[g^ | x] = g? I think this is a more realistic model, and the same proofs should go through.


Review 3

Summary and Contributions: This paper considers non-smooth constrained convex optimization which provides an algorithm with optimal first-order oracle complexity, eps^{-2}, and projection oracle complexity only eps^{-1}, which improves over existing methods which required eps^{-2} projection oracle calls. Similarly, when a linear minimization oracle is available instead of a projection oracle, their method requires only eps^{-2} LMO calls rather than the previously known eps^{-4}. ------- update after author response -------- After reading the author's response, I maintain my score and still recommend accepting the paper. I agree with Reviewer 1 that Lemma 1 should be stated more carefully, but I do not see this as a fatal flaw.

Strengths: This is a nice paper which clearly presents the problem and its solution. Reducing the number of projection oracle calls is very important and can lead to much faster optimization algorithms (e.g. for nuclear norm constraints), and the method is novel to the best of my knowledge. The ideas used for the algorithm and its proof are quite straightforward, which is another strength of the paper.

Weaknesses: This is a strong paper with few weaknesses. The main theoretical limitation seems to be the requirement that the function/first order oracle be available outside the constraint set. While I generally agree with the authors that this is often reasonable, I think it would be useful to think about (and comment on) cases in which this might fail. It seems that two things might go wrong here: either (1) the Lipschitz constant could explode or (2) the function could be undefined. This might be an issue if the objective involves an entropic regularizer, or something like that? Again, I don't think this is a huge issue, but thinking through these issues would be good. This is especially true since I suspect that it is probably not necessary to have \mathbb{X} be so much larger than \mathcal{X}, I suspect that a much smaller expansion of the constraint set should still be workable.

Correctness: To the best of my knowledge the claims and methods are correct.

Clarity: Yes, the paper is clearly written. There are a couple of minor typos: Line 114: "sest's" -> set's Line 183-184: It should be "...PO calls are NOT required..." Pseudocode 1.14: u_{k,t} is not defined/used elsewhere. I think this should just be u_t?

Relation to Prior Work: Yes, the paper includes a thorough comparison with prior work and clearly explains its additional contribution.

Reproducibility: Yes

Additional Feedback:

[Author Response · NeurIPS 2020]

We sincerely thank all the reviewers for their detailed comments and queries, and give clarifications and answers below.

**Reviewer 1:** We will revise according to all the comments on typos, clarity, and rigor of the mathematical writing.
First, we focus on the correctness of Lemma 1, and then address the others comments.
**Correctness.** We agree with the reviewer that Lemma 1 is not precise, as we (mistakenly) did not include the precise
range of $x$ in the statements. We provide here the precise version of Lem.1, where the differences are colored in BLUE:
**Lemma 1.** *For a closed convex set $\mathbb{X}$, a convex proper l.s.c. function $f : \mathbb{X} \to \mathbb{R} \cup \{\infty\}$ and $\lambda > 0$ define $f_\lambda : \mathbb{R}^d \to \mathbb{R}$*
*as $f_\lambda(x) := \min_{x' \in \mathbb{X}} f(x') + \frac{1}{2\lambda} \|x - x'\|^2$ and $\hat{x}_\lambda(x) := \operatorname{argmin}_{x' \in \mathbb{X}} f(x') + \frac{1}{2\lambda} \|x - x'\|^2$. Then for any $x \in \mathbb{X}$:*
*(a) $\hat{x}_\lambda(x)$ is unique and $f(\hat{x}_\lambda(x)) \leq f_\lambda(x) \leq f(x)$.*
*(b) $f_\lambda$ is convex, differentiable, $1/\lambda$-smooth and $\nabla f_\lambda(x) = (1/\lambda)(x - \hat{x}_\lambda(x))$, and,*
*(c) if $f$ is G-Lipschitz continuous, then, $\|\hat{x}_\lambda(x) - x\| \leq G\lambda$, and $f(x) \leq f_\lambda(x) + G^2\lambda/2$.*

This version of Lemma 1 is $(i)$ sufficient for proving our main results (lines 577, 622, 691, 705) and $(ii)$ correct. Since
the reviewer's counter example uses $x \notin \mathbb{X}$, it does not contradict Lemma 1(c). We now provide a full proof below.
*Proof (brief due to page limit).* Denote $\phi_{\lambda,x}(x') := f(x') + (1/2\lambda)\|x - x'\|^2$. Note that $\phi_{\lambda,x}(\cdot)$ is a $1/\lambda$-strongly
convex function and $f_\lambda(x) = \min_{x' \in \mathbb{X}} \phi_{\lambda,x}(x')$.
(a) Then $f(\hat{x}_\lambda(x)) \leq \phi_{\lambda,x}(\hat{x}_\lambda(x)) = \min_{x' \in \mathbb{X}} \phi_{\lambda,x}(x') = f_\lambda(x) \leq \phi_{\lambda,x}(x) = f(x)$ and the uniqueness of $\hat{x}_\lambda(x)$
follows from the strong convexity of $\phi_{\lambda,x}(\cdot)$ and the fact that $f$ is a proper convex function.
(b) Let $x \in \mathbb{R}^d$ and $g_x := (x - \hat{x}_\lambda(x))/\lambda$. By $1/\lambda$ strong convexity of $\phi_{\lambda,x}(x')$ and $\hat{x}_\lambda(x) = \operatorname{argmin}_{x' \in \mathbb{X}} \phi_{\lambda,x}(x')$,
we have for any $x' \in \mathbb{X}$ that $\phi_{\lambda,x}(x') \geq \phi_{\lambda,x}(\hat{x}_\lambda(x)) + \|x' - \hat{x}_\lambda(x)\|^2/2\lambda$, which simplifies to $f(x') \geq f(\hat{x}_\lambda(x)) +$
$\langle g_x, x' - \hat{x}_\lambda(x)\rangle$. Using this, for any $x, y \in \mathbb{R}^d$ we get

$$f_\lambda(y) - f_\lambda(x) = f(\hat{x}_\lambda(y)) - f(\hat{x}_\lambda(x)) + (\|\hat{x}_\lambda(y) - y\|^2 - \|\hat{x}_\lambda(x) - x\|^2)/2\lambda$$
$$\geq \langle g_x, \hat{x}_\lambda(y) - \hat{x}_\lambda(x)\rangle + \lambda/2(\|g_y\|^2 - \|g_x\|^2) = \langle g_x, y - x\rangle + \lambda/2\|g_x - g_y\|^2 \quad (1)$$

Instantiating the above for $y \leftarrow x$, $x \leftarrow y$ we also get $f_\lambda(y) - f_\lambda(x) \leq \langle g_y, y - x\rangle - \lambda/2\|g_x - g_y\|^2$. Combining
these two inequalities

$$0 \leq \lambda/2\|g_y - g_x\|^2 \leq f_\lambda(y) - f_\lambda(x) - \langle g_x, y - x\rangle \leq -\lambda/2\|g_y - g_x\|^2 + \langle g_y - g_x, y - x\rangle \leq \|y - x\|^2/2\lambda \quad (2)$$

This implies that $\lim_{y \to x}(f_\lambda(y) - f_\lambda(x) - \langle g_x, y - x\rangle)/\|y - x\| = 0$. Thus $f_\lambda$ is Frechet differentiable with gradient
$\nabla f_\lambda(x) = g_x = (x - \hat{x}_\lambda(x))/\lambda$. The above inequality also implies $f_\lambda$ is convex and $1/\lambda$-smooth.
(c) Let $x \in \mathbb{X}$. Using $1/\lambda$-strong convexity of $\phi_{\lambda,x}$ and $\hat{x}_\lambda(x) \in \operatorname{argmin}_{x' \in \mathbb{X}} \phi_{\lambda,x}(x')$, and $G$-Lipschitzness of $f$,

$$\|x - \hat{x}_\lambda(x)\|^2/2\lambda \leq \phi_{\lambda,x}(x) - \phi_{\lambda,x}(\hat{x}_\lambda(x)) = f(x) - f_\lambda(x) = f(x) - f(\hat{x}_\lambda(x)) - \|x - \hat{x}_\lambda(x)\|^2/2\lambda$$
$$\leq G\|\hat{x}_\lambda(x) - x\| - \|x - \hat{x}_\lambda(x)\|^2/2\lambda \leq G^2\lambda/2 . \quad \square$$

We say line 557: "for simplicity...$\mathbb{X}$ is the whole vector space". This was an assumption made, in the context of Sec. A.1,
for ease of exposition of the failed attempt at a PO efficient algorithm (Algo. APGD).
**Experimental verification.** As suggested, we compared the projection-free methods using a higher-dimensional
($d = 50,176$) ImageNet dataset in the same low-rank SVM problem. For achieving an optimality gap of $0.02$,
Randomized-FW[52] used $34717/264$ FO/LMO calls and our MOLES used $4004/241$ FO/LMO calls. We will add
detailed simulation results including sensitivity analysis in the next revision. We agree that our algorithms have more
parameters and hence harder to tune than most baselines. Overcoming this is an important direction of future research.

**Reviewer 2** We agree that the reviewer's definition of the stochastic subgradient oracle is more appropriate. We
modified the manuscript according to the additional comments.

**Reviewer 3** The two properties we need of the superset $\mathbb{X} \supseteq \mathcal{X}$ are that (a) it is easy to project onto $\mathbb{X}$ and (b) $f$ is
$G$-Lipschitz on $\mathbb{X}$. In our paper, we choose $\mathbb{X}$ to be a Euclidean ball (which is easy to project to) but any other choice of
$\mathbb{X}$ which satisfies the above properties works just as well. One choice for this Euclidean ball is $B(x_0, D_{\mathcal{X}})$, where $x_0$ is
the initial point and $D_{\mathcal{X}}$ is the diameter of $\mathcal{X}$, instead of the ball of radius $2R$ we currently use.

As mentioned by R3, even if $f$ is $G$-Lipschitz inside the constraint $\mathcal{X}$, it could (i) blow up or (ii) be undefined just
outside of $\mathcal{X}$. Thus an $\mathbb{X}$ satisfying our requirements may not exist. In our experiments, we do not explicitly project onto
$\mathbb{X}$ (line 3.16) but still observed that $\|x_k - x_k'\| = O(G\lambda)$ and small, which implies that the iterates $x_k'$ are close to
$\mathcal{X}$. This hints that we may only need Lipschitzness over a much smaller set $\mathcal{X} + B(0, O(G\lambda))$, but we do know how to
prove this yet. Theoretically, we can work around the issue (ii) above by minimizing the convex extension $f_{\mathcal{X}} : \mathbb{R}^d \to \mathbb{R}$
of the function $f$ from the set $\mathcal{X}$, defined as $f_{\mathcal{X}}(x') := \max_{x \in \mathcal{X}} \max_{g \in \partial f(x)} f(x) + \langle g, x' - x\rangle$. The extension $f_{\mathcal{X}}$
has the same value as $f$ inside $\mathcal{X}$ and is $G$-Lipschitz everywhere. Therefore the following minimization problems
are equivalent: $\min_{x \in \mathcal{X}} f(x)$ and $\min_{x \in \mathcal{X}} f_{\mathcal{X}}(x')$. However, it is not clear if we can even estimate/approximate the
gradients of $f_{\mathcal{X}}$ efficiently. We could not find any relevant prior work and leave this question for future work. We
modified the manuscript according to the additional comments.

[Meta-Review · NeurIPS 2020]

All reviewers agreed that this paper made a nice contribution to NeurIPS and recommend acceptance. It considers a novel perspective on constrained non-smooth optimization (by proposing algorithms that reduce the number of projection / linear-minimization oracle calls), improving the previous state-of-the-art rates (removing d^1/4 for the projection oracles and sqrt(d) for LMO over previous best rates, obtaining an optimal rate for LMO and answering an open problem dating from 1993 [83]!). While R1 initially expressed concerns about the writing and the technical correctness of Lemma 1, the reviewers agreed in discussion that this was properly addressed in the rebuttal. The authors should implement these changes in the camera ready version of the paper, as discussed in their rebuttal (clarity, etc.; also include the LMO new empirical results).